# A distributed soil moisture, temperature and infiltrometer dataset for permeable pavements and green spaces

Axel Schaffitel[1], Tobias Schuetz[2], Markus Weiler[1]

[1]Faculty of Environment and Natural Resources, University of Freiburg, Freiburg i. Br., 79098, Germany
[2]Faculty of Regional and Environmental Sciences, University of Trier, Germany

*Correspondence to*: Axel Schaffitel (axel.schaffitel@posteo.de)

**Abstract.** Knowledge on water and energy fluxes is a key for urban planning and design. Nevertheless, hydrological data for urban environments is sparse and as a result, many processes are still poorly understood and thus inadequately represented within models. We contribute to reduce this shortcoming by providing a dataset, which includes time series of soil moisture and soil temperature measured underneath 18 different permeable pavements (PPs) and 4 urban greenspaces located within the city of Freiburg (Germany). Time series were recorded with a high temporal resolution of 10 min with a total of 65 individual soil moisture sensors and cover a measuring period of 2 entire years (Nov. 2016 – Oct. 2018). The recorded time series contain valuable information on the soil hydrological behavior of PPs and demonstrate the effect of surface properties and surrounding urban structures on soil temperatures. In addition, we performed double-ring infiltration experiments, which in combination with the soil moisture measurements yielded soil hydrological parameters for the PPs including porosity, field capacity and infiltration capacity. We present this unique dataset, which is a valuable source of information for studying urban water and energy cycles. We encourage its usage in various ways e.g. for model calibration and validation purposes, to study thermal regimes of cities and to derive urban water and energy fluxes. The dataset is freely available at the FreiDok plus data repository at https://freidok.uni-freiburg.de/data/151573 and https://doi.org/10.6094/UNIFR/151573 (Schaffitel et al., 2019)

# 1 Introduction

Knowledge of urban water and energy fluxes is a key for urban planning and design. Although, there are various urban hydrological (see Elliott and Trowsdale, 2007 for a review on urban hydrological and drainage models) and energy balance models (see Grimmond et al., 2010 for an overview), there is only limited data available for validating and calibrating those models (Litvak et al., 2017; Salvadore et al., 2015; Schirmer et al., 2013). As a result, many processes remain unclear and are poorly represented within models (Salvadore et al., 2015). To overcome this shortcoming, Vereecken et al. (2015) emphasize the possibilities of new measurement technologies.

Urbanization leads to profound changes of water and energy cycles (Oke, 1988; Shuster et al., 2005). Impacts of altered energy fluxes include the formation urban heat islands (UHIs) in the atmosphere, but also in the subsurface of cities (Oke et al., 2017). Impacts on the water balance include increased surface runoff volumes (Fletcher et al., 2013; Shuster et al., 2005) at the expense of soil infiltration (Cristiano et al., 2017; Salvadore et al., 2015; Schirmer et al., 2013) and evapotranspiration (Fletcher et al., 2013; Grimmond and Oke, 1991). One possibility to mitigate the hydrological impacts of urbanization is to replace impermeable surface covers with permeable pavements (PPs). Although, their usage is restricted mainly to parking spaces, pedestrian roads and roads with low traffic volumes, PPs can cover great parts of cities (Winston et al., 2016). Positive effects include the reduction of surface runoff volumes, reduction in peak flows as well as increases in evaporation and groundwater recharge rates (Andersen et al., 1999; Fassman and Blackbourn, 2010; Park et al., 2014; Scholz and Grabowiecki, 2007; Timm et al., 2018).

Soil moisture ($\theta$) is of major importance for understanding water and energy fluxes in terrestrial systems (Eagleson, 1978; Lahoz and De Lannoy, 2014; Trenberth and Asrar, 2014). One example is the partitioning of rainfall into surface runoff and infiltration, which depends decisively on the state of the soil storage (Brocca et al., 2008). Although, this partitioning is of special interest for urban stormwater management, the effect of $\theta$ on the hydrologic performance of PPs is still under debate. While some authors reported antecedent moisture conditions to effect the hydrologic performance of PPs (Brown and Borst, 2015; Fassman and Blackbourn, 2010), other authors found a limited effect of antecedent moisture conditions on the hydrologic performance of PPs (Guo et al., 2018). We anticipate that this debate will benefit from continuous soil moisture measurements below PPs. So far, such measurements were used to analyze the applicability of time-domain reflectometry in coarse structured soils (Ekblad and Isacsson, 2007; Stander et al., 2013), to derive water fluxes (Ragab et al., 2003), for calibration of plot scale soil hydrologic models (Kodešová et al., 2014; Turco et al., 2017) and to study clogging dynamics of PPs (Razzaghmanesh and Borst, 2018). As far as known to the authors, a comprehensive analysis of $\theta$-dynamics beneath PPs did not take place so far. In contrast to soil moisture measurements, various authors have performed infiltration experiments which were used e.g. to derive the infiltration capacity of PPs (Illgen, 2009), for studying clogging dynamics (Borgwardt, 2006; Lucke and Beecham, 2011) and to analyze the effect of road maintenance on infiltration rates (Winston et al., 2016).

Soil temperature ($T_{soil}$) contains important information on surface and subsurface thermal regimes. Analyzing them is of particular interest, as subsurface temperatures effect the groundwater quality and the gas exchange of soils (Oke et al., 2017), but also have implications for the use of geothermal energy (Zhu et al., 2010). Furthermore, $T_{soil}$ can be used to derive the ground heat flux (GHF) (Kimball et al., 1976). Thereby, the GHF of urban areas is of special interest, as it is the main driver for the formation of subsurface UHIs (Menberg et al., 2013) and fundamental for closing the urban energy balance (Grimmond and Oke, 1991; Roberts et al., 2006). Nevertheless, values for the urban GHF are sparse and calculations often depend on parameter assumptions leading to high uncertainties. We are convinced that $T_{soil}$–observations will improve the prediction of the urban energy balance and may reveal new findings on the formation and magnitude of subsurface UHIs.

According to Salvadore et al. (2015), there is a special need for measurements within urban soils. As far as we know, there is neither $\theta$ nor $T_{soil}$ data freely available for urban environments. Here we provide a unique dataset comprising of $\theta$ and $T_{soil}$ measured below PPs and green spaces located within an urban environment. Furthermore, we performed infiltration experiments which together with the $\theta$-time series were used to derive soil hydrological parameters for PPs. As porous surface covers (porous asphalt, porous concrete and porous paving stones) play a minor role within the study area, they are not part of dataset. Instead, data for PPs is limited to surface covers consisting of impermeable pavers separated by permeable joints (e.g. interlocking concrete pavers, cobblestones, grass pavers). We are convinced that the provided dataset is of great value for studying urban water and energy fluxes and encourage its usage within the fields of urban hydrology and urban climatology.

## 2. Study area

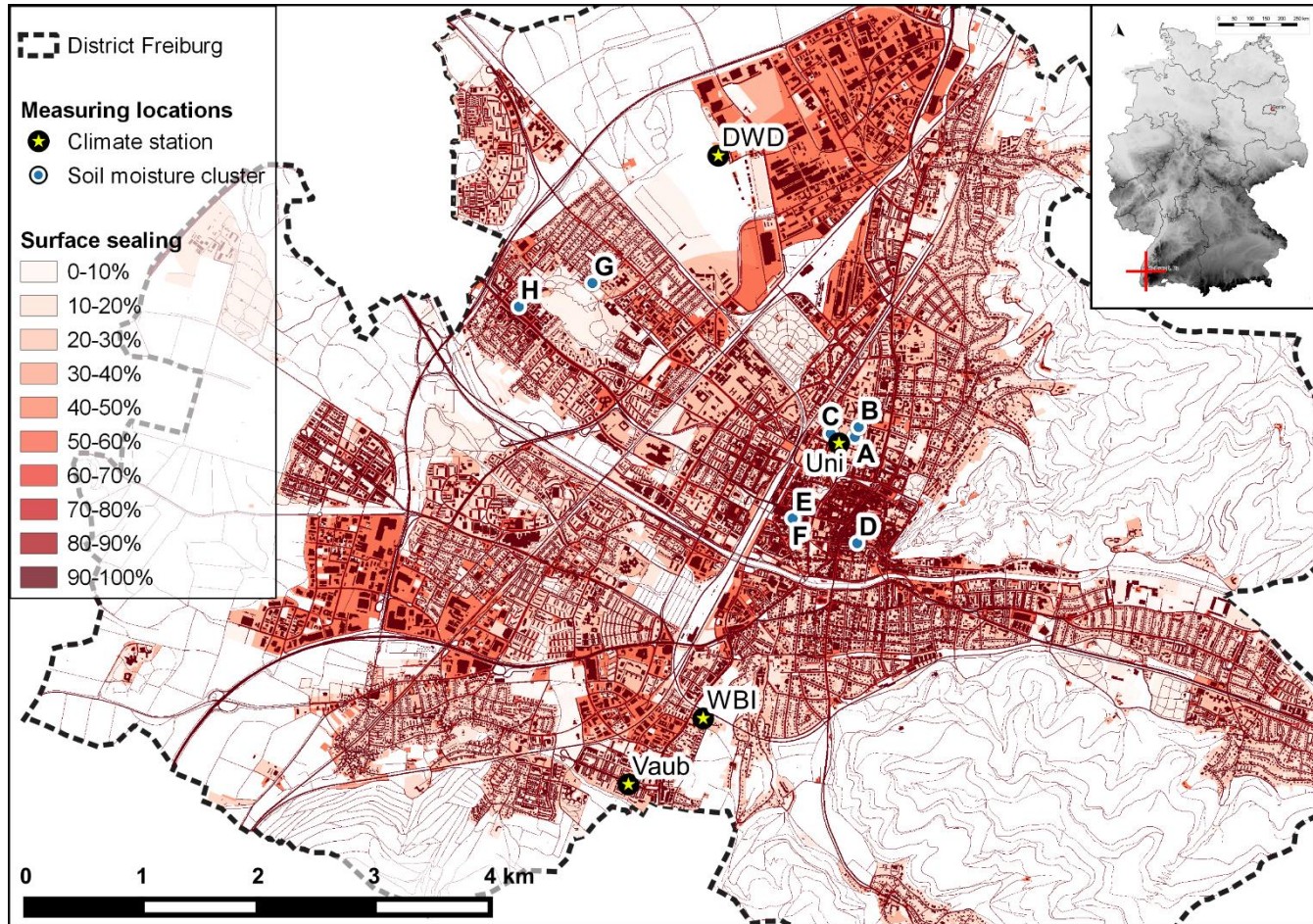

**Figure 1: Location of the soil moisture clusters and the climate stations within the city of Freiburg. Colors indicate the degree of surface sealing** (LUBW, 2012)**. Geodata of Germany in the upper right box originates from the Federal Agency for Cartography and Geodesy of Germany, while geodata of Freiburg was provided by the city of Freiburg.**

Soil moisture measurements were carried out in the city of Freiburg i. Br., which lies in the southwest of Germany (Figure 1). Natural soils found within the district of Freiburg are dominated by Cambisols and Luvisols at surrounding hillsides, whereas Fluvisols and Gleysols prevail within the valleys (soil map 1:50.000 federal state authority for Geology and Natural Resources of Baden-Württemberg). Urbanization impacts natural soils and strongly alters their properties (Wessolek, 2008). Therefore, large parts of natural soils within cities have been transformed into Technosols and Anthrosols (Kodešová et al., 2014). The study area is located in a temperate climate with a mean annual precipitation (*P*) of 894 mm and an annual mean air temperature of 11.5°C (evaluated for the period 1996-2015 for the climate station of the German Weather Service (DWD)). Rainfall is seasonally uneven distributed with more rainfall occurring during summer due to convective storms. The two hydrological

years studied (Nov. 2016 – Oct. 2018) were characterized by around 200 mm/year less *P* compared to the long-term average (Figure 2).

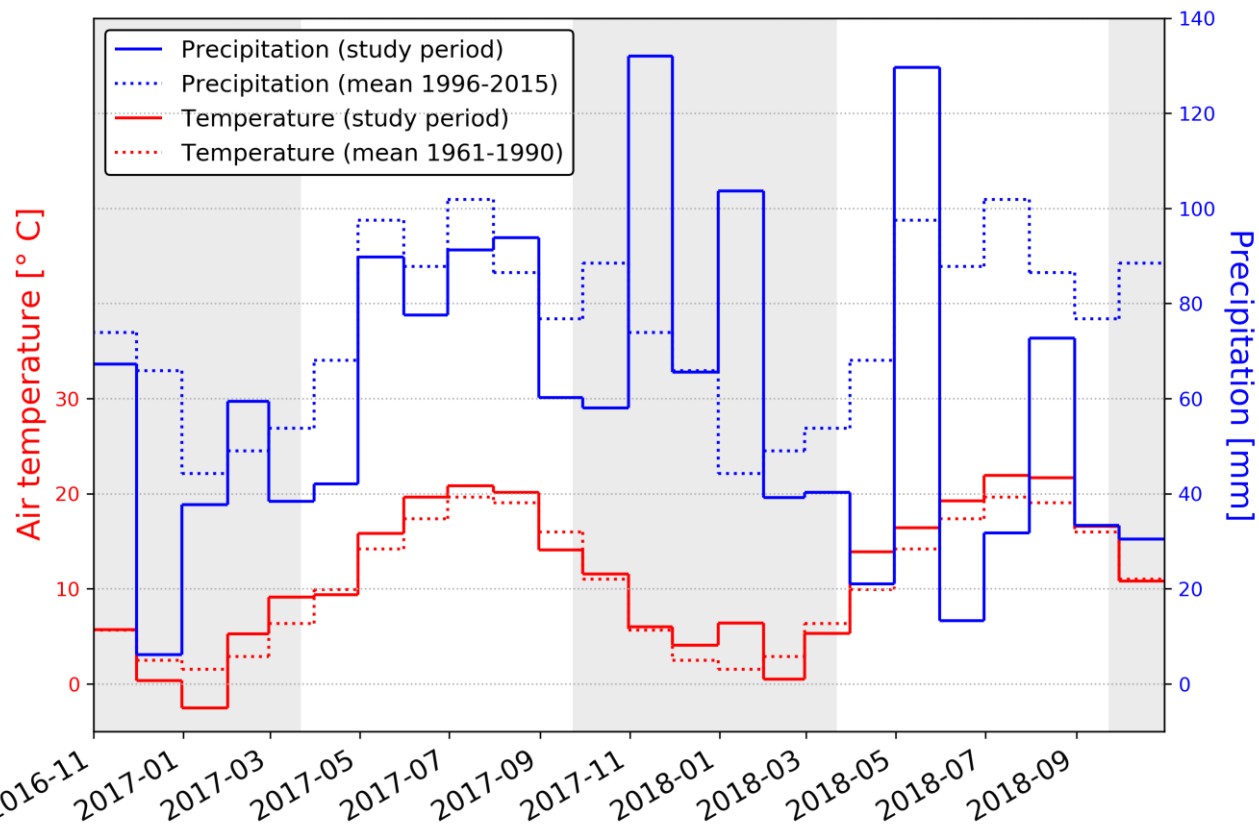

5    **Figure 2: Monthly precipitation sum and mean air temperature recoded at the climate station DWD within the study period compared to long-term mean values. The light grey background indicates the winter half year.**

## 3. Material and methods

### 3.1 Permeable pavements

According to German regulations, PPs must be designed to fully infiltrate a 10-minute lasting rainfall with a return period of 5 years. To ensure this, the surface must show an infiltration capacity of at least 97.2 mm/h (Borgwardt, 2001). Furthermore, the saturated hydraulic conductivity of underlying soils should exceed 19.4 mm/h. In case of lower permeability, the installation of underdrains is required. Figure 3 shows the typical layers of a permeable pavement which is built in accordance to national regulations (Borgwardt, 2001; FGSV, 2012). This structure may be adapted locally, depending e.g. on constructional requirements and on the permeability of underlying soils. Paving stones, bedding, base and subbase layers consist of technical substrates which are installed during construction works. Their main function is to absorb and distribute pressures equally and to drain infiltrating water rapidly with the goal to ensure the bearing capacity and the frost-resistance of the pavement layer. In most cases, the subbase layer lies directly above heavily compacted natural soils. Only in special cases, a subgrade is needed below the subbase layer (Borgwardt, 2001). PPs and their underlying layers belong to the soil class of Technosols.

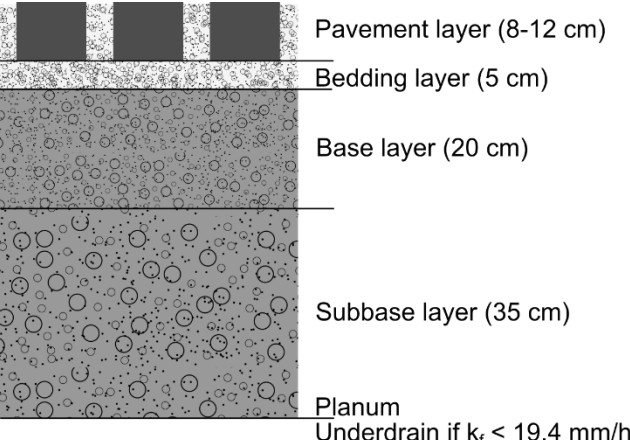

**Figure 3: Layers of a typical permeable pavement build in accordance to national regulations.**

## 3.2 Climate data

There are four different climate stations (Figure 1) available within the study area (see Sect. data availability for URLs of the individual climate stations). Data of the individual climate stations differ in resolution, documentation, provided variables and vicinity to soil moisture clusters. Therefore, data users should select the climate data in dependence of their specific purpose.

Over the study period, only the time series of the DWD and WBI climate stations are free from data gaps. Since the DWD climate station is operated according to the guidelines of the World Meteorological Organization, the available documentation is best for this station and the measured climate variables should be unbiased by urban effects. In contrast, the remaining three climate stations are located in close vicinity to urban structures and therefore are affected by the urban climate. We expect the climatic input of the soil moisture measurements to be best represented by the WBI climate station, as it is the only urban

climate station that is free from data gaps. This climate station is operated by the State Viniculture Institute Freiburg. Online available data is limited to an hourly temporal resolution, while data with a 10 min temporal resolution was provided upon request by the Center for Agricultural Technology Augustenberg (belonging to the Ministry of the Environment, Rural Affairs and Consumer Protection of the state Baden-Württemberg). In order to facilitate the use of high-resolution climate data for the WBI climate station and to ensure its long-term availability, we asked for the permission to include this data in our data

repository. For event separation, rain events were defined as rainy periods exceeding a minimum of 0.5 mm and separation took place when no rainfall was recorded for at least 2 h. In order to account for the spatial variability of urban rainfall (Cristiano et al., 2017) the event separation was based on the $P$ data recorded at both gap-free climate stations (DWD and WBI). In this way, 302 individual events were separated for the study period (Nov. 2016 until Oct. 2018). Reference crop evapotranspiration ($et_0$) is a key variable for most hydrological studies and was calculated for the WBI climate station by using

the Pennman-Monteith equation and the parametrization provided by Allen et al. (1998). The time step recommended for the calculation of $et_0$ is one day (Allen et al., 1998). Since a high temporal resolution might be desirable for further data users, we decided to provide $et_0$ also with an hourly temporal resolution.

## 3.3 Measurement network

The soil measurement locations (hereinafter called plots) are organized in clusters with each cluster comprising several different surfaces (see Figure 1 for the location of the clusters). Hemispherical photos were taken at each cluster and are included within the data repository. Using those photos allows to calculate the sun path for each day of the year (see Figure 9 for an example) and therefore enables for a time-dependent quantification of potential shading and insolation. To calculate the sun path, we used the software package RayMan (Matzarakis et al., 2007). Furthermore, we analyzed the fraction of different

urban structures within a 5 m and 10 m radius around each cluster by means of a GIS analysis. The results of this analysis are shown in Appendix A (Table 5) and are further included within the data repository (file metaClusters.txt). All studied PPs are located on either parking lots, pedestrian roads or residential roads with low traffic volumes. On each plot, infiltration

experiments were performed and soil moisture sensors (SMT100, Truebner GmbH, Mannheim, Germany) were installed in 2-4 depths below the ground surface. Installation of the SMT100 sensors took place during two field campaigns in March and June 2016. For the installation, the existing pavement was removed and sensors were inserted into the undisturbed profile wall under undisturbed pavements. In order to avoid water ponding on the sensors and to minimize the disturbance of vertical vapor fluxes, they were installed horizontally with the narrow side in vertical direction. Afterwards, the plots were refilled with the original base and bedding material. Finally, municipal construction workers restored the paving layer professionally. Due to the high compaction of soils and the presence of coarse aggregates, the installation of the SMT100 sensors often was challenging. Therefore, we decided for some plots to insert sensors not into the profile wall but instead install them within the excavated hole, which afterwards was refilled successively with bedding material (material used for the construction of the bedding layer). Therefore, soil hydraulic properties between the refilling and the original soil material should be comparable within the bedding layer. Soils found within the underlying layers were characterized by a strong heterogeneity and the hydraulic properties of the refilling should lie within the variability occurring within those layers. Each cluster was equipped with an individual data logger accommodated in a small manhole. Variables measured by the SMT100 include the apparent dielectric permittivity ($\varepsilon_a$) and the soil temperature ($T_{soil}$) and were recorded by the data logger with a time interval of 10 min. Figure 4 shows the installation of the soil moisture sensors at the plots of cluster D as well as the small manhole used for accommodating the logger.

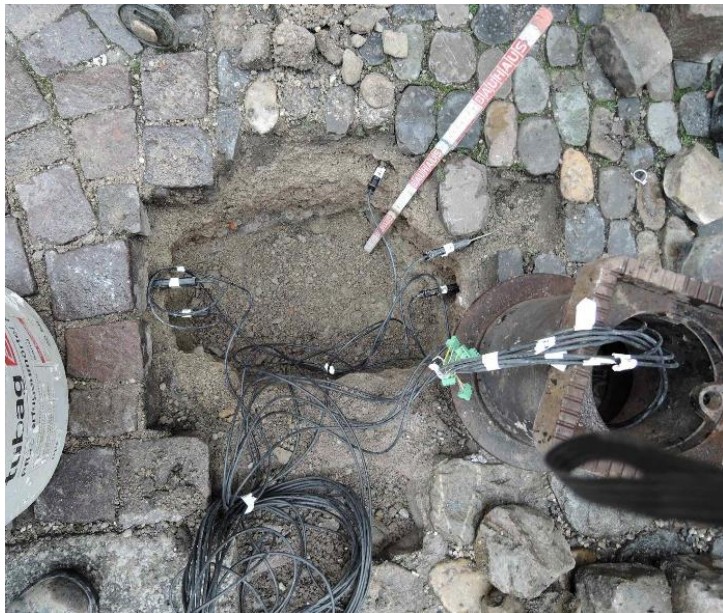

Figure 4: Installation of the soil moisture sensors at the plots of cluster D and the small manhole used for accommodating the data logger.

During the installation, soil samples were taken and the characteristics of the joints were analyzed. Recorded joint characteristics include joint material, joint condition, mean joint depth and joint vegetal cover. The mean joint depth was calculated from 10 individual measurements performed with a high precession slide gauge with a resolution of 0.1 mm. Joint conditions were classified into 4 categories ranging from 0 (very good condition) to 3 (very bad condition). This was achieved by a combination of visual categorization and finger testing of the joint material (not performed at PPs with mortar filled joints). Thereby, higher amount of fines were attributed to higher clogging and consequently to worse joint condition. Similarly, joint vegetal cover (consisting of tread-resistant plant communities and mosses) was classified visually into 4 categories ranging from 0 (no vegetation) to 3 (fully covered by vegetation). Finally, image analysis tools were used to determine the proportion of joints and the degree of surface sealing. For this purpose, a 1 m x 1 m frame was mounted on the surfaces and pictures were taken. The images were fist corrected for geometric distortion by using an adapted version of the algorithm described in Weiler and Flühler (2004). Afterwards, paving stones were digitized manually and their areal proportion was calculated. These digitized images capture the small-scale variability of the PP surfaces and are included within the data repository. Table 1 shows the plots of the measurement network and their characteristics.

**Table 1: Characteristics of the plots**

| Name | A1 | A2 | A3 | A4 | B1 | B2 | C1 | C2 | C3 | D1 | D2 | D3 | E1 | E2 | F1 | F2 | G1 | G2 | H1 | H2 | H3 | H4 |
|---|---|---|---|---|---|---|---|---|---|---|---|---|---|---|---|---|---|---|---|---|---|---|
| Cluster | A | A | A | A | B | B | C | C | C | D | D | D | E | E | F | F | G | G | H | H | H | H |
| Surface type[1] | lawn | CCP | GP | CCP | NP | lawn | DCP | DCP | bushes | NP | NP | NP | NP | NP | NP | NP | DCP | CCP | DCP | CCP | CCP | lawn |
| Sealing degree [%][2] | 0 | 93 | 54 | 92 | 83 | 0 | 89 | 73 | 0 | 80 | 81 | 86 | 76 | 75 | 76 | 78 | 86 | 97 | 85 | 94 | 92 | 0 |
| Build [year] | - | 1999 | 1999 | 1999 | 2011 | - | 2004 | 2004 | - | 1998 | 1998 | 1998 | 1999 | 1999 | 2014 | 2014 | 1987 | 1987 | 1985 | 1985 | 1985 | - |
| Height paving layer [cm] | - | 8 | 10 | 8 | 8 | - | 8 | 10 | - | 10 | 8 | 12 | 10 | 10 | 10 | 10 | 10 | 10 | 10 | 10 | 10 | - |
| Joint condition [cat.] | - | 3 | 2 | 3 | 0 | - | 1 | 1 | - | 1 | 1 | 1 | 2 | 2 | 0 | 0 | 3 | 3 | 2 | 2 | 2 | - |
| Joint material | - | sand | vegetated gravel-sand | sand | gravel-sand | - | gravel-sand | gravel-sand | - | mortar | gravel-sand | gravel-sand | mortar | mortar | mortar | mortar | sand | sand | sand | sand | sand | - |
| Joint vegetal cover [cat.] | - | 2 | 3 | 3 | 0 | - | 2 | 3 | - | 0 | 0 | 0 | 3 | 2 | 1 | 0 | 2 | 0 | 2 | 2 | 2 | - |
| Mean joint depth [mm] | - | 2 | 9 | 2 | 11 | - | 2 | 14 | - | 10 | 12 | 24 | 10 | 11 | 4 | 3 | 4 | 5 | 5 | 3 | 6 | - |
| Installation method[3] | profile wall | profile wall | profile wall | profile wall | filled | profile wall | filled | filled | profile wall | profile wall | profile wall | profile wall | filled | filled | filled | filled | filled | filled | filled | filled | filled | profile wall |

1: Abbreviations of surface types follow the classification of Timm et al. (2018) with CCP: classical concrete paving stones; DCP: designed concrete paving stones; GP: grass pavers; NP: natural paving stones.

2: The sealing degree equals 100% minus the joint proportion [%].

3: Sensors were installed with 2 different methods. Either they were installed in the undisturbed profile wall (profile wall) or in the excavated hole which then was filled from the top (filled)

4: The profiles show the 2 soil components with the biggest share. Minor components are not visualized. All profiles are presented to a uniform depth of 45 cm, although not all plots were excavated till this depth. In this case, the lowest layer encountered was extrapolated until the depth of 45 cm for illustration purposes. In case of installation method "filled", the depicted soil layers show the encountered soil material during excavation instead of the bedding material used for refilling.

### 3.4 Soil moisture and temperature measurements

Soil moisture and $T_{soil}$-measurements were carried out in 10 min intervals by the SMT100 sensors, which belong to the family of time domain transmission sensors (Qu et al., 2013). The SMT100 sensor operates by inducing a steep pulse to a closed transmission line of known length (Bogena et al., 2017). Thereby, the travel time of the pulse along the transmission line depends on the dielectric properties of the surrounding soil and forms the basis for water content measurements (Topp et al., 1980). Instead of measuring the travel time directly, the SMT100 measures an oscillation frequency, which is directly related to travel time (Qu et al., 2013). According to Bogena et al. (2017) it accounts around 150 MHz in water and 300 Hz in air. The sensor internally transforms the measured oscillation frequency into the dielectric permittivity ($\varepsilon_c$) (Bogena et al., 2017). Thereby, the error of the permittivity measurement is estimated from the results of Bogena et al. (2017) who tested 701 SMT100 sensors in reference liquids with known dielectric properties. Their results indicate that the error of the permittivity measurement is below 1.5. Afterwards, the Topp equation (Topp et al., 1980) is applied to derive the volumetric water content ($\theta$). According to the manufacturer, the resolution of the SMT100 is at least 0.1 vol.% for $\theta$ and 0.01°C for $T_{soil}$. The accuracy in determining absolute values is 3 vol.% for $\theta$ (without soil specific calibration of the sensors) and between 0.2°C and 0.4°C for $T_{soil}$. Variables measured by the SMT100 represent average values over the entire sensor length of 10 cm. Measurements below PPs should therefore integrate substrates lying below joints as well as substrates lying below paving stones.

Temperature affects electromagnetic soil moisture measurements in various ways (Kapilaratne and Lu, 2017; Qu et al., 2013; Wraith and Or, 1999). Thereby, we observed strong temperature oscillations beneath the PPs with amplitudes reaching 20°C/day and extreme values ranging between -6°C and 47°C. These strong thermal variations affected the soil moisture measurements notably and called for a temperature correction of $\theta$. We therefore applied 3 different correction methods which are described in detail in the following.

First, we used the Complex Refraction Index Model (CRIM) model of Roth et al. (1990) (eq. 1) to consider the temperature-dependency of liquid water permittivity ($\varepsilon_w$) (eq. 2) (Handbook of Physics and Chemistry, 1986 as cited by Roth et al., 1990). The effect of measurement errors (dielectric permittivity and temperature) and parameter uncertainties (porosity, permittivity of the gaseous and solid phase) on $\theta$ calculated by the CRIM equation was quantified by Roth et al. (1990) to not exceed 1.3 vol.%.

$$\theta = \frac{\varepsilon_c^\alpha - \varepsilon_s^\alpha - \phi(\varepsilon_a^\alpha - \varepsilon_s^\alpha)}{\varepsilon_w^\alpha - \varepsilon_a^\alpha} \qquad \text{(eq. 1)}$$

$$\epsilon_w(T_W) = 78.54 \left[1 - 4.579 * 10^{-3} (T_{soil} - 25) + 1.19 * 10^{-5} (T_W - 25) - 2.8 * 10^{-8}(T_W - 25)^3\right] \qquad \text{(eq. 2)}$$

Parameter values for eq. 1 were taken from Roth et al. (1990) with a geometry factor ($\alpha$) of 0.46 and a temperature independent permittivity of the gaseous ($\varepsilon_a$) and the solid phase ($\varepsilon_s$) of 1.0 and 3.9 respectively. The porosity ($\phi$) is derived in Sect. 5 to account 0.37 (median of all plots). In eq. 2, we used $T_{soil}$ as a proxy for the water temperature ($T_w$).

Afterwards, the calculated $\theta$-values were aggregated to hourly values and the data-driven temperature correction method of Kapilaratne and Lu (2017) was applied. Although, this reduced temperature-induced $\theta$-oscillations noticeably, they were still present during periods with low saturation. Therefore, we used linear interpolation between daily minima for further correcting $\theta$. This linear interpolation was applied only for dry periods and thereby started 24 h after the ending of antecedent rain events. For users who want to apply a different temperature correction scheme, the original $\varepsilon_c$ data is included within the dataset. Since
dielectric properties of frozen and liquid water differ from that of liquid water, times with frozen soils were removed from the $\theta$-time series (but not from the $\varepsilon_c$ time series). This was achieved by removing all times during which $T_{soil}$-values were below 0.5°C at any point in the profile or frozen joints were observed visually. However, users interested in $\theta$ during those times may use an adapted version of the CRIM model, which enables to consider the dielectric properties of frozen water (examples for such an adapted version of the CRIM model are used e.g. by Demand et al. (2019) and by Roth and Boike (2001)). One source
of error in $\theta$ may arise from salt spreading during winter road maintenance. This potentially leads to seasonal changes of soil salinity which in turn effects $\varepsilon_c$ (Ekblad and Isacsson, 2007). As electrical conductivity is not measured by the SMT100 sensor, it was neither possible to quantify the impact, nor to correct for. Hence, care should be taken when analyzing winter data of $\theta$.

Values reported in literature for the saturated water content ($\theta_s$) of base and bedding layers of PPs range between 20-45 vol.%
(Brunetti et al., 2016; Illgen et al., 2007; Kodešová et al., 2014). As it is one of the main functions of PPs to drain infiltrating water rapidly to the subsurface, we expect $\theta$ to recede fast after ending infiltration and to hardly ever reach $\theta_s$. In contrast, at PPs with low permeability of underlying soils and missing or malfunctioning underdrains, we expect recorded $\theta$-values to reach $\theta_s$ and afterwards to recede slowly. Based on this, we classified the drainage behavior of the PPs into the categories "free drainage" and "restricted drainage". Therefore, we analyzed the empirical frequency distribution of $\theta$ recorded during rain
events and calculated the mode of this distribution (see Figure 7 for an example), which can serve as a measure for the classification. However, using the mode as a single threshold would neglect the effect of soil properties on the frequency distribution of soil moisture. Therefore, we further incorporated a visual analysis of the recorded $\theta$-time series to classify the drainage behavior of the PPs. In case time series were characterized by a fast rise and recession of soil moisture (flashy behavior), a low mode occurred and the drainage behavior was classified as "free drainage". In contrast, a high mode was
calculated for time series where $\theta$ reached $\theta_s$ frequently during rainfall. Those time series were classified as "restricted drainage".

To analyze the hydrologic behavior of different urban surface covers, we compared the $\theta$-dynamics recoded at different plot categories. Therefore, we calculated a mean $\theta$-time series for the categories "vegetated", "PPs with free drainage" and "PPs with restricted drainage". Thereby, times with data-gaps at any of the sensors were excluded, as gaps cause jumps in the calculated mean. Furthermore, the plots of cluster C are not included in the comparison, as the measurements at this cluster

5    ended in Jan 2018 due to construction works.

### 3.5 Infiltration experiments

Infiltration experiments under falling head conditions were performed in spring 2016 on the plots. Therefore, we used double-ring infiltrometers with diameters of 320 mm and 556 mm of the inner and the outer ring, respectively. Bentonite was used to

10   form a tight seal between the pavement layer and the infiltrometer. Figure 5 shows the performance of the infiltration experiments at the plots of cluster D.

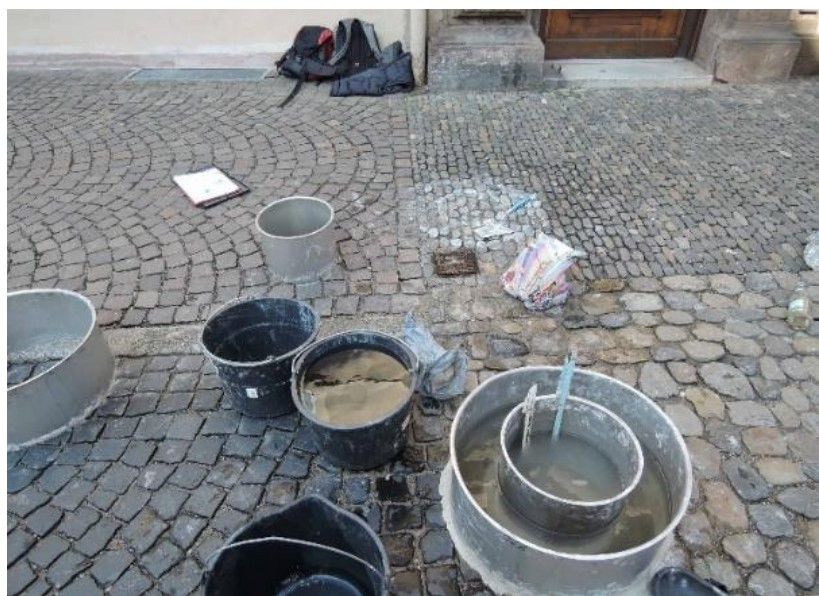

**Figure 5: Infiltration experiments at the plots of cluster D**

Ponding depth was around 12.5 cm at the beginning of the experiments and water tables were recorded visually with a temporal resolution of 1 min. The reading accuracy of the visual observations is approx. 0.5 mm. However, measurement errors should mainly cancel out over the infiltration course, since a cumulative quantity was measured. Refilling occurred when water tables fall below 0.5 cm. Experiments were performed over 45 min except on plots with high infiltration rates, since for most of the

20   plots, the water supply was limited to 200 L. As expected, infiltration rates were higher at the beginning of the experiment

(initial infiltration rates) and decreased over time until they reached a constant rate (final infiltration rate). The main cause for this decline is the decrease of matrix suction gradients with the proceeding of the infiltration front (Hillel, 1998).

### 3.6 Derivation of soil hydrological parameters

Depending on the drainage behavior, soil moisture measurements were used to either derive $\theta_s$ (for PPs with restricted drainage) or to derive $\theta_{fc}$ (for PPs with free drainage). For PPs with free drainage, we used the median of all $\theta$-values recorded in the period 48-72 h after rain events to estimate $\theta_{fc}$. To ensure that soils were initially wetted above $\theta_{fc}$, only events with a precipitation sum of at least 10 mm were considered for assessing $\theta_{fc}$. At PPs with restricted drainage, saturation was reached frequently during rainfall. We used the 0.99 quantile of $\theta$ measured during rain events to assess $\theta_s$ of these plots, which is assumed to equal $\phi$. As $\theta$ calculated with the CRIM model (eq. 1) depends on $\phi$, we used $\theta$ calculated with the $\phi$-independent Topp equation (Topp et al., 1980) to derive $\theta_s$.

The Philip infiltration model (Philip, 1957) (eq. 3) was used to describe the cumulative infiltration ($I$) measured during the infiltration experiments.

$$I = St^{\frac{1}{2}} + At \quad \text{(eq. 3)}$$

where $S$ is the sorptivity, $t$ is time and $A$ is a parameter representing the theoretical minimum of the infiltration rate (approached asymptotically for large values of $t$), which is defined as endinfiltration rate. The two model parameters $S$ and $A$, as well as their confidence intervals were determined inversely, by fitting the model to the observed $I$ using the least-square optimization algorithm implemented in the Python module SciPy (function optimize.curve_fit). After determining the parameters, we used the fitted Philip model to calculate the mean infiltration rate over a period of 5 h, which we defined as infiltration capacity ($i_{cap}$). The duration of 5 h represents the median duration of the 302 separated rainfall events within the study period. In addition, we assessed the initial infiltration rate ($i_{start}$) by calculating the mean infiltration rate over the first 10 min of infiltration. By calculating $i_{cap}$ and $i_{start}$ with parameter values located at the boundary of their confidence intervals, we determined the uncertainties of $i_{cap}$ and $i_{start}$. Finally, the depression storage of joints ($S_{joints}$) was assessed by multiplying the mean joint depth by the joint proportion (Table 1).

## 4. Data

### 4.1 Soil moisture

After applying the CRIM model, temperature effects were still present in the $\theta$-time series. They manifest in form of diurnal soil moisture oscillations, which are characterized by increasing $\theta$ with rising $T_{soil}$. Their occurrence contrasts the findings of Bogena et al. (2017), who showed that $\theta$ derived by the CRIM model was free from temperature effects. A possible explanation for the different findings may be given by the effect of bound water, which gets partially released with increasing $T_{soil}$ (Or and Wraith, 1999). While the effect of bound water may be negligible at high $\theta$, it might play an important role at low $\theta$. In contrast to the study of Bogena et al. (2017), $\theta$ measured within this study reaches much lower values. In our dataset, temperature effects are most pronounced during those periods with low $\theta$, which supports the assumption that the observed temperature effects are caused by bound water. However, we used two additional methods to correct for temperature effects, which together turned out to be effective in removing the diurnal soil moisture oscillations.

Most of the studied PPs indicate free drainage, as $\theta$ recedes fast after the end of a precipitation event. In contrast, 6 of the 18 studied PPs (plots of cluster H, B and C) revealed restricted drainage, as $\theta$ often reached a plateau during rainfall which persisted even after the end of a precipitation event. At the plots of cluster H restricted drainage was apparent at all sensors, while it was mainly limited to lower layers at the plots of clusters B and C. Figure 6 shows the depth-dependent comparison between $\theta$ measured at vegetated plots, PPs with free drainage and PPs with restricted drainage, while Table 2 gives an overview on the number of sensors included within each category. The mean and the standard deviation of all sensors within a category is shown in Appendix A (Table 6).

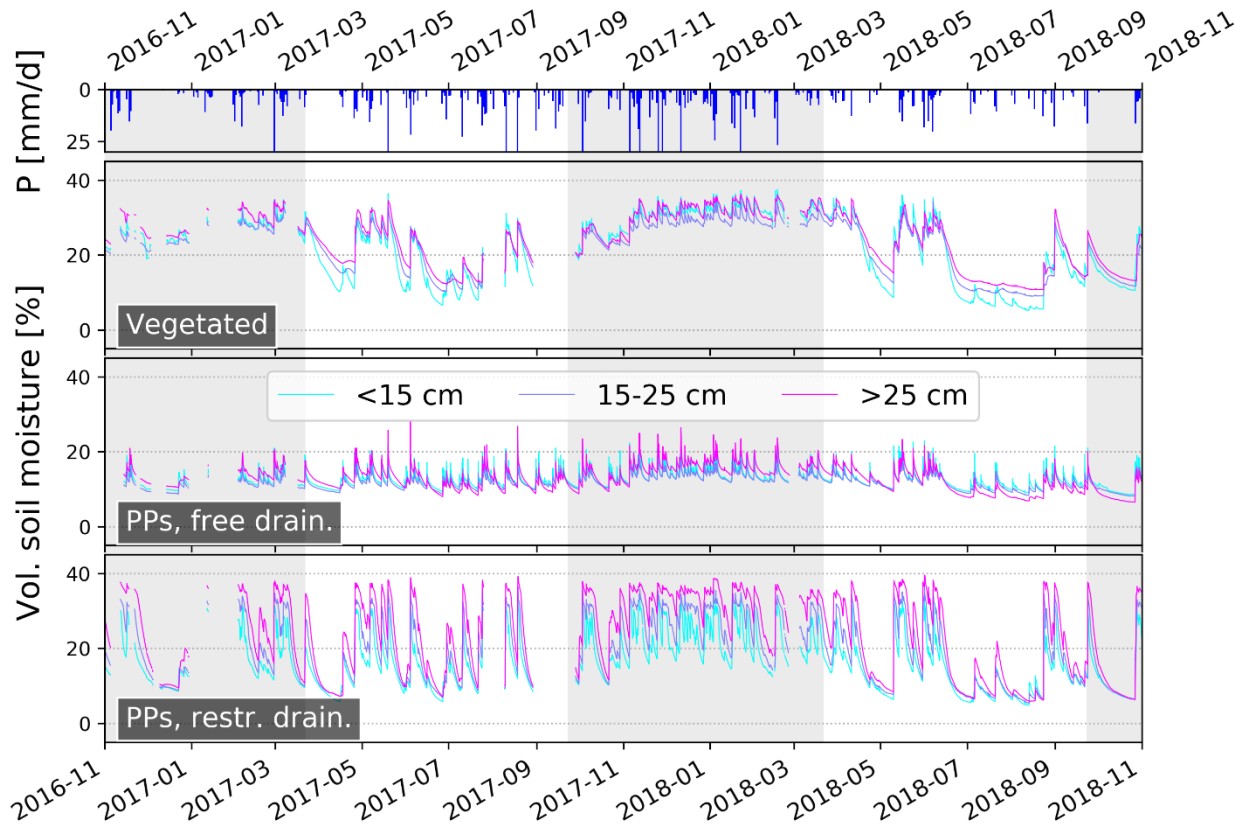

**Figure 6: Mean $\theta$ measured underneath vegetated surfaces (top), PPs with free drainage (middle) and PPs with restricted drainage (bottom). The grey background indicates the winter half year.**

5    **Table 2: Number of sensors included within each category Figure 6**

| | Depth below ground surface | | |
|---|---|---|---|
| | <15 cm | 15-25 cm | >25 cm |
| **Vegetated** | 3 | 3 | 3 |
| **PPs with free drainage** | 13 | 11 | 10 |
| **PPs with restricted drainage** | 4 | 4 | 5 |

Soil moisture  time series recorded underneath PPs differ substantially from those recorded underneath vegetated surfaces. At the vegetated plots, a seasonal pattern is apparent, which comprises a wet state during the winter half year. During the wet state, $\theta$ is similar in all depths while during summer, the upper layers dry out faster and stronger. A further depth-dependency

10    is apparent, as the dampening and delay of the $\theta$-rise increases with depth. In contrast, the category "PPs with free drainage" show hardly any seasonal and depth-dependent differences. Rise and recession of $\theta$ occurs very fast and reveals a high "flashiness" of the soil water storage. Finally, PPs with restricted drainage show depth-dependent differences, as plateaus in $\theta$

are more persistent and appear more frequent at greater depths. Furthermore, a wet state can be identified, as $\theta$ recedes seldom in winter.

One of the main functions of PPs is to drain infiltrating water rapidly to the subsurface and hence to avoid temporary saturation. The flashiness of the soil water storage observed at PPs with free drainage indicates that these plots fulfil this function. In contrast, the plateaus apparent in $\theta$ recorded at PPs with restricted drainage indicate that soils were often saturated. Possible effects are the formation of saturation overland flow as well as impacts on the bearing capacity and the frost-resistance. This shows the relevancy of a high permeability of underlying soils, and is in contrast to the findings of numerical studies which reported a limited influence of deeper soil layers on the hydrologic performance of PPs (Brunetti et al., 2016; Illgen et al., 2007). The difference between plots with free drainage and plots with restricted drainage gets further apparent in the empirical frequency distribution of $\theta$-values recorded during rain events (Figure 7).

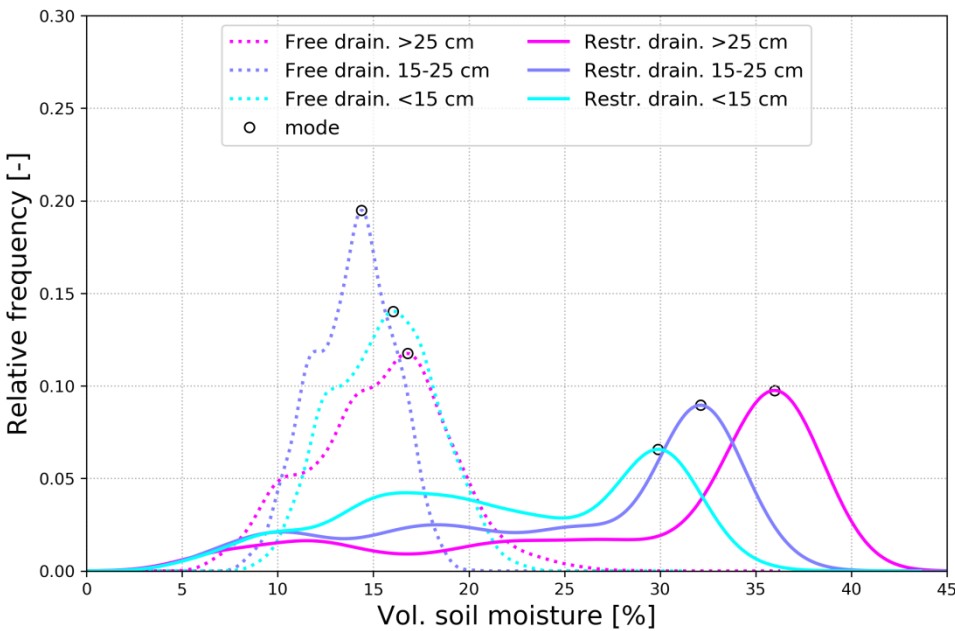

**Figure 7: Density distribution of $\theta$-values measured at PPs during rain events depending on drainage behavior and sensor depth. Depicted densities are mean values of all sensors within each category.**

During rain events, $\theta$-values are much lower on PPs with free drainage and values above 25 vol.% are hardly ever reached. In contrast, $\theta$-values measured on PPs with restricted drainage are frequently above 25 vol.% during rain events. As bedding and base layers are technical substrates with defined hydrologic properties, the porosities of all PPs should be similar. The

frequency distribution recorded at PPs with free drainage indicates that these plots hardly ever reached saturation during the study period. Therefore, infiltration excess overland flow is the only process leading to surface runoff on PPs with free drainage.

## 4.2 Soil temperatures ($T_{soil}$)

The recorded $T_{soil}$-values show periodic variations at the diurnal and at the annual scale. Figure 8 shows the mean of the measured $T_{soil}$-values in different depths for PPs and vegetated surfaces, while Table 7 of Appendix A shows the mean and the standard deviation of all sensors within each category.

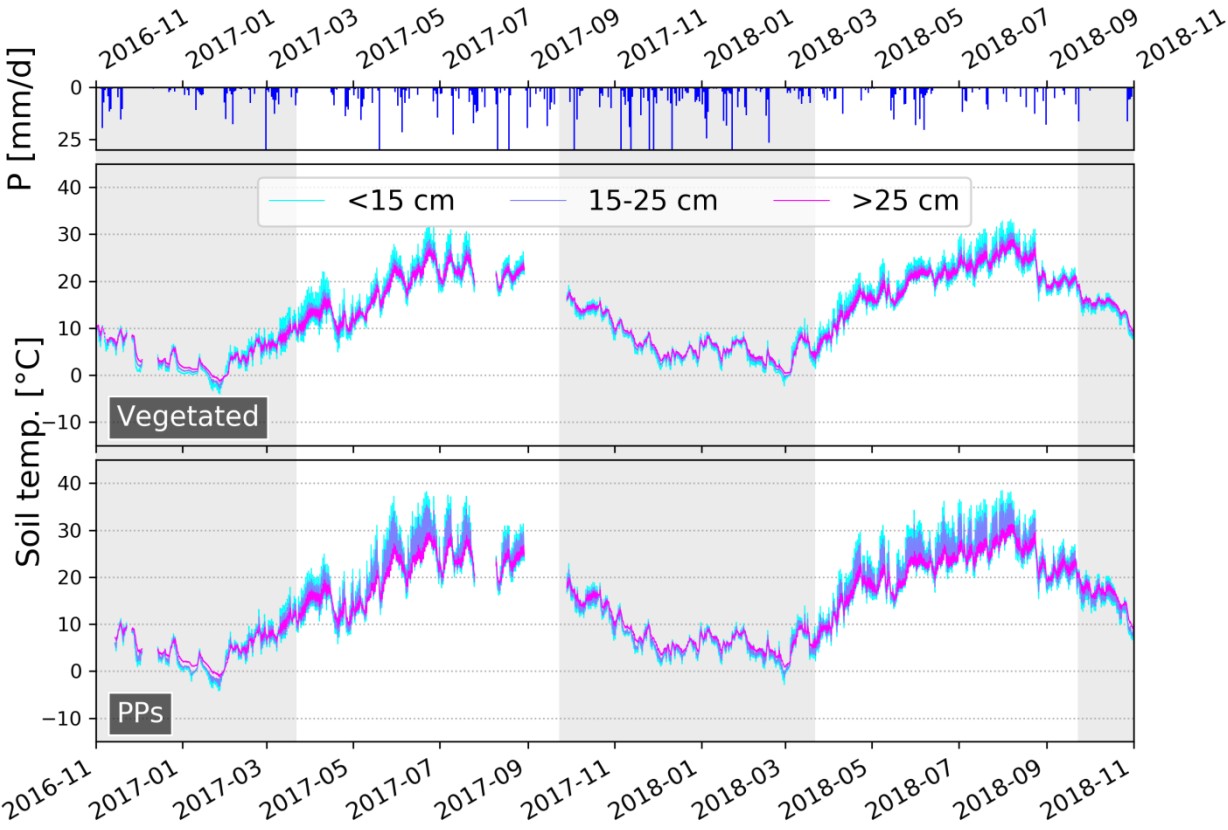

**Figure 8: Mean soil temperatures measured in different depths for vegetated (top) and paved surfaces (bottom). The grey background indicates the winter half year.**

Both diurnal and annual amplitudes are more pronounced on PPs than on vegetated plots. Differences are mostly pronounced
10 during the summer half year, when PPs undergo a much stronger diurnal heating. This is also reflected in the recorded maxima $T_{soil}$-values, which account 37°C on vegetated and 47°C on PPs. The heterogeneity of the urban environment causes a high spatial variability in the climatic input and therefore strongly effects $T_{soil}$. Of all clusters, highest $T_{soil}$-values occurred at cluster D, which is located close to a south-orientated facade of an east-west orientated urban canyon within the city center. In contrast, maxima soil $T_{soil}$-values recorded at the shaded Cluster A were around 17°C lower. Figure 9 exemplarily shows the sun path

during summer solstice for cluster A and for cluster D. During this day, cluster D potentially receives irradiation from around 07:00-18:00 o'clock, while cluster A receives markedly less irradiation.

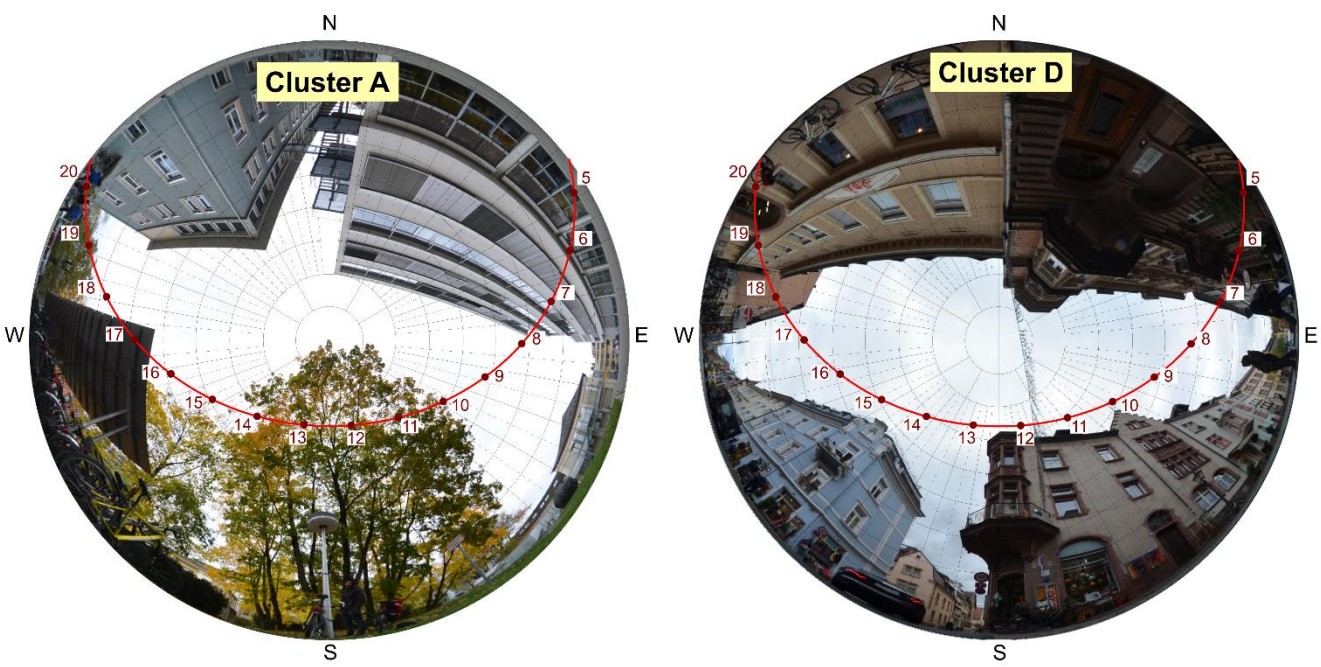

**Figure 9: Sun path at cluster A and cluster D during summer solstice, calculated from hemispherical photos by using RayMan (Matzarakis et al., 2007). The points indicate the position of the sun at the full hour (numbers), while the red line indicates the sun path over the day.**

Another characteristic of cluster A is the similarity between $T_{soil}$-values measured at the vegetated and at the paved plots. In contrast, pronounced differences between vegetated and paved plots occurred at cluster H, which is only partially shaded by surrounding objects. Figure 10 exemplarily shows the diurnal course of temperature depth-profiles measured at the vegetated and at a paved plot of cluster H on August 6[th], 2018. During this day, the diurnal $T_{soil}$-amplitude is strongest at the uppermost sensors and is dampened with depth. Compared to the vegetated plot, the diurnal amplitude in 30 cm depth, is more pronounced on the paved plot. This indicates that temperature oscillations penetrate to greater depths, which is expected to be mainly caused by the occurrence of higher surface temperatures on the paved plot. The measured temperature-depth profiles offer the opportunity to derive ground surface temperatures, the penetration depth of periodic temperature variations and can further be used to calculate the ground heat flux.

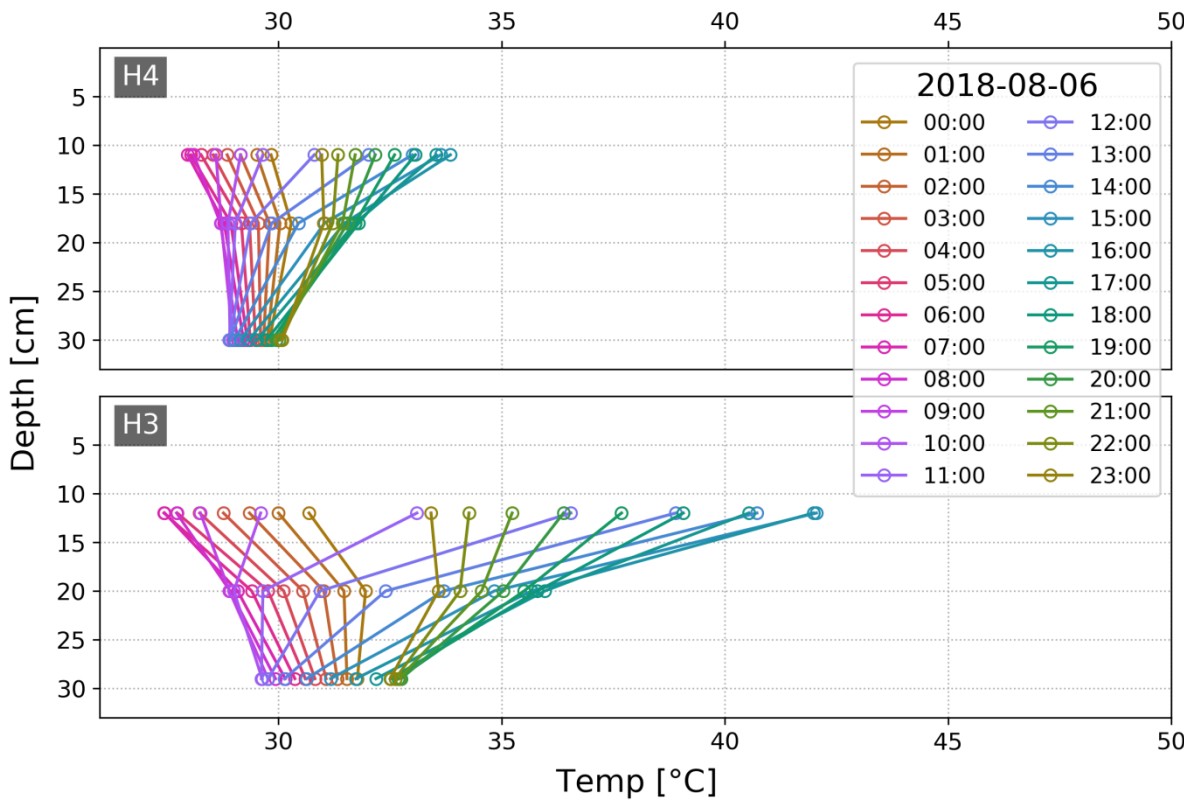

**Figure 10: Temperature-depth profiles measured at cluster H on August 6th, 2018 at the vegetated plot (top) and the paved plot (bottom)**

## 4.3 Infiltration experiments

Infiltration experiments were performed at all PPs except at the plots E2, F2 and H3. Thereby plots E1/E2 as well as plots F1/F2 were constructed during the same field campaigns (same age, equal soil material used for base and bedding layers), have the same proportion of joints and are exposed to similar microclimatical conditions. Figure 11 shows the data recorded during
10   the infiltration experiments together with the fitted Philip infiltration model (eq. 3).

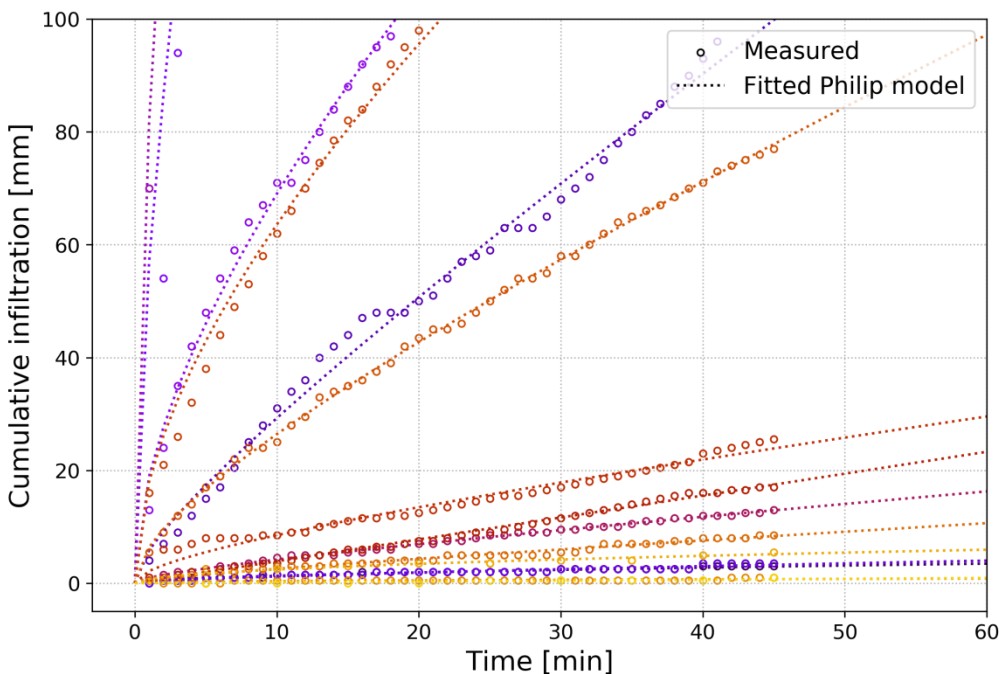

**Figure 11: Measured infiltration and simulation with the Philip infiltration model.**

## 5. Derived soil hydrological parameters and transferability of the data

The measurements were used to derive soil hydrological parameters of the PPs. Depending on the drainage behavior of the PPs, the soil moisture measurements were either used to derive $\theta_{fc}$ or $\theta_s$. Data of the infiltration experiments was used to derive the parameters $S$ and $A$ of the Philip infiltration model, which in turn were used to derive $i_{cap}$ and $i_{start}$. The depression storage of joints ($S_{joints}$) was assessed from the mean joint depth and the joint proportion. Table 3 shows the derived hydrological parameters for the PPs.

**Table 3: Derived hydrologic parameters for the PPs. The parameters $\theta_{fc}$ and $\theta_s$ are shown for three different depth ranges, while the remaining parameters are provided independent of depth. The $A$- parameter is considered as endinfiltration rate.**

| | $\theta_{fc}$ [vol.%] | | | $\theta_s$ [vol.%] | | | $S$ [mm/min$^{1/2}$] | $A$ [mm/min] | $i_{cap}$ [mm/h] | $i_{start}$ [mm/h] | $S_{joints}$ [mm] |
|---|---|---|---|---|---|---|---|---|---|---|---|
| | <15 cm | 15-25 cm | >25 cm | 15-25 cm | <15 cm | >25 cm | | | | | |
| A2 | 9.91 | 18.11 | 29.47 | - | - | - | 0.39 ± 0.02 | 0.01 ± 0.00 | 1.79 ± 0.36 | 7.86 ± 0.75 | 0.14 |
| A3 | 11.88 | - | 14.94 | - | - | - | 4.23 ± 0.42 | 1.59 ± 0.08 | 109.92 ± 6.27 | 175.57 ± 12.80 | 5.04 |
| A4 | 14.64 | 10.94 | 12.31 | - | - | - | 0.29 ± 0.04 | 0.03 ± 0.01 | 2.73 ± 0.57 | 7.27 ± 1.18 | 0.16 |
| B1 | - | - | - | 36.72 | 36.76 | 39.37 | 47.04 ± 1.33 | 9.85 ± 0.26 | 754.25 ± 20 | 1483.86 ± 41 | 1.87 |
| C1 | - | - | - | 38.53 | 36.51 | 40.2 | 17.60 ± 0.40 | 1.34 ± 0.08 | 141.19 ± 6.21 | 414.10 ± 12.42 | 0.22 |
| C2 | - | - | - | 36.02 | 38.35 | 36.81 | 66.55 ± 5.65 | 15.05 ± 2.37 | 1133.29 ± 166 | 2165.48 ± 250 | 3.78 |
| D1 | 10.2 | 15.7 | - | - | - | - | 0.79 ± 0.03 | 0.17 ± 0.01 | 12.89 ± 0.47 | 25.13 ± 0.98 | 2 |
| D2 | 14.54 | 13.7 | - | - | - | - | 0.00 ± 0.06 | 0.39 ± 0.01 | 23.29 ± 0.88 | 23.29 ± 1.84 | 2.28 |
| D3 | 14.94 | 16.28 | - | - | - | - | 1.91 ± 0.17 | 0.25 ± 0.03 | 21.37 ± 2.48 | 50.99 ± 5.16 | 3.36 |
| E1 | 12.58 | 10.63 | 11.88 | - | - | - | 17.04 ± 0.45 | 0.97 ± 0.08 | 117.14 ± 6.41 | 381.39 ± 13.34 | 2.4 |
| E2 | 18.15 | 11.55 | 9.18 | - | - | - | - | - | - | - | 2.75 |
| F1 | 11.33 | 7.11 | 11.9 | - | - | - | 5.46 ± 0.11 | 0.92 ± 0.02 | 73.85 ± 1.57 | 158.50 ± 3.26 | 0.96 |
| F2 | 11.62 | 8.95 | 10.7 | - | - | - | - | - | - | - | 0.66 |
| G1 | 11.7 | 13.9 | 14.19 | - | - | - | 0.41 ± 0.05 | 0.12 ± 0.01 | 8.90 ± 0.73 | 15.21 ± 1.52 | 0.56 |
| G2 | 10.94 | 10.62 | 15.43 | - | - | - | 0.11 ± 0.02 | 0.00 ± 0.00 | 0.37 ± 0.36 | 2.02 ± 0.74 | 0.15 |
| H1 | - | - | - | 36.46 | 36.22 | 37.3 | 0.77 ± 0.14 | 0.00 ± 0.03 | 2.65 ± 2.00 | 14.53 ± 4.20 | 0.75 |
| H2 | - | - | - | 36.04 | 36.6 | 35.94 | 0.00 ± 0.07 | 0.02 ± 0.01 | 0.93 ± 1.00 | 0.93 ± 2.09 | 0.18 |
| H3 | - | - | - | 36.34 | 35.25 | 36.01 | - | - | - | - | - |
| Median | 11.79 | 11.55 | 12.31 | 36.4 | 36.56 | 37.06 | 0.78 ± 0.09 | 0.21 ± 0.02 | 17.13 ± 1.28 | 24.21 ± 2.68 | 0.96 |
| Standard deviation | 2.41 | 3.34 | 5.98 | 0.94 | 1.01 | 1.78 | 19.88 | 4.37 | 329.53 | 1.49 | 2.41 |

The calculated $\theta_{fc}$-values range between 7 vol.% and 29 vol.%, with values above 20 vol.% occurring only at plot A2 in 40 cm depths. The median of the calculated $\theta_{fc}$-values is similar within all depth ranges and the median of all sensors accounts 11.89 vol.% with a standard deviation of 3.89 vol.%. The median of the obtained $\theta_s$-values accounts 36.46 vol.% for sensors

located in bedding layer and 36.59 vol.% for sensors located in the base layers, while the standard deviation accounts 0.92 vol.% and 1.38 vol.%, respectively. Obtained $\theta_s$-values are within the range reported in literature. Illgen (2009) obtained $\theta_s$-values of 45% (base layer) and 37% (bedding layer) by calibration of the HYDRUS-2D model with lysimeter data. Similarly, Brunetti et al. (2016) used a particle swarm optimization algorithm to calibrate HYDRUS-1D. They obtained $\theta_s$-values of 30% and 20% for the bedding and the base layer respectively (single porosity model). Multistep outflow experiments were performed by Kodešová et al. (2014) leading to $\theta_s$-values of 42% and 43% for the bedding and base layer respectively. As bedding and base layers are technical substrates introduced during the construction of PPs, we assume the derived values for $\theta_s$ and $\theta_{fc}$ to be representative also for other PPs.

Values obtained for $i_{cap}$ range between 0.37 mm/h and 1133 mm/h. According to national regulations, PPs must show a minimum $i_{cap}$ of 97.2 mm/h (Borgwardt, 2001). The derived $i_{cap}$-values show that only 5 of the 15 tested PPs satisfy this requirement. Values below 2 mm/h were obtained for the plots A2, G2 and H2, which are characterized by a low proportion of joints and a bad joint condition. Various authors showed that main clogging takes place within the first years after the installation of PPs (Boogaard et al., 2014; Borgwardt, 2006; Lucke and Beecham, 2011). As the age of all plots (except the plots of cluster F) lie between 14-20 years, all plots are expected to be affected by clogging. Fassman and Blackbourn (2010) highlight the role of adjacent land use on clogging. The 3 plots with lowest $i_{cap}$ are located in close vicinity to urban greenspaces and trees, which might explain the low $i_{cap}$-values of those plots. Infiltration characteristics of the 2 PPs with mortar filled joints (E1 and F1) are similar, despite the differences in their age and joint condition. The mortar filled joints of plot E1 were classified as "bad", while for plot F1, they were classified as "very good". Despite the worse joint condition, plot E1 showed a higher $i_{cap}$ than plot F1. Since the bad joint condition of plot E1 comes along with broken and crumbly joint mortar, the occurrence of cracks may compensate the clogging of the mortar matrix and therefore may explain the high $i_{cap}$ measured on plot E1.

The presented dataset poses a valuable source of information for the urban environment. However, the pronounced heterogeneity in urban surface coverage and in urban soil composition aggravates the transferability of the dataset to other urban sites. However, since soil layers underneath PPs consist of technical substrates with defined hydrological properties, the soil moisture patterns underneath PPs should be similar. Hence, the observed patterns should be transferable to other urban sites. This is also the case for the parameters derived from soil moisture measurements ($\theta_s$ and $\theta_{fc}$). In contrast, various authors highlighted the variability of the infiltration capacity of PPs, which decisively depends on the state of joint clogging (e.g. Illgen (2009)). Therefore, the transferability of the infiltrometer data is limited.

## 6. Data availability

The dataset is available at the FreiDok plus data repository at https://freidok.uni-freiburg.de/data/151573 and https://doi.org/10.6094/UNIFR/151573 (Schaffitel et al., 2019) and contains time series of $\theta$, $T_{soil}$, $\varepsilon_c$ and climate data measured at the WBI station. Furthermore, it comprises time series of $et_0$ (calculated for the WBI climate station) and event classification.

Soil moisture time series were obtained by applying the temperature correction procedures described in Sect. 3.4. and are available with an hourly temporal resolution. For users who need $\theta$ with a higher temporal resolution, or want to apply different temperature correction schemes, $\varepsilon_c$ is provided with a 10 min temporal resolution. Soil temperature is also provided with a temporal resolution of 10 min, while the event classification is available with hourly resolution. Time series of $et_0$ exist with two different temporal resolutions (daily and an hourly). Furthermore, the original data of the infiltration experiments is

provided. Cluster, sensor and plot specific meta-information is supplied in three separate files. All hydrologic parameters derived within this study are included in these files. Furthermore, the data repository contains hemispherical photos of each cluster and binary images of all PPs. A readme file contains information on the organization of all folders and files. Online sources for the data of the climate stations depicted in Figure 1 are listed in Table 4 (Note that for the DWD climate station, metadata is only provided when selecting hourly resolution data).

**Table 4: Sources of climate data for the study area**

| Station | Temporal resolution | Source |
|---------|---------------------|--------|
| DWD | up to 1 min (free download) | https://opendata.dwd.de/climate_environment/CDC/observations_germany/climate/ (Station-ID: 01443) |
| WBI | 1 h (free download) ; 10 min (on request) | http://www.wetter-bw.de/Internet/AM/NotesBwAM.nsf/bwweb/fa31cce8b142f059c1257ca7003c9ed1?OpenDocument |
| Vaub | 10 min (free download); 1 min (on request) | Link "Vauban Meteo Station"on http://hydro.uni-freiburg.de/strt-en?set_language=en |
| Uni | 1 min (free download) | https://weather.uni-freiburg.de/ |

## 7. Summary and conclusions

We provide a unique dataset of soil moisture and soil temperatures measured within an urban environment. So far, only few studies conducted measurements within urban soils and there is a need for more observations (Salvadore et al., 2015). The analysis of the soil moisture contents revealed important information on the behavior of permeable pavements within the urban environment. Recorded soil moisture contents revealed restricted drainage to occur on 6 of the studied 18 permeable

pavements, leading to temporary saturation in the subsurface of this plots. The observed water saturation potentially affects the bearing capacity and the frost-resistance of the pavements, but may also lead to the formation of saturation overland flow. This contradicts the results of previous simulation studies which revealed a limited effect of underlying soil layers on the

hydrologic performance of PPs (Brunetti et al., 2016; Illgen et al., 2007). We used the times with saturation to derive the saturated water content and the porosity of these plots. The median of the derived saturated water contents accounts 37 vol.%. The remaining 12 permeable pavements revealed free drainage and showed fast soil moisture recession after rainfall events. This flashiness causes that saturation is hardly ever reached and that neither pronounced seasonal nor depth-dependent differences appear in soil water contents. The infiltration capacity of these pavements is restricted only by joint properties and surface runoff generation is limited to infiltration excess overland flow. Measured soil moisture contents were used to derive the field capacity of base and bedding layers. The median of the obtained field capacities accounts 12 vol.% and is similar within depths. Since base and bedding layers are technical substrates with defined hydrological properties, the derived saturated water contents and field capacities should be representative for other permeable pavements. In addition to soil moisture and temperature measurements, infiltration experiments were performed on 15 of the permeable paved plots and used to derive their infiltration capacity. The results showed that only 5 of the studied permeable pavements fulfilled national regulations. Lowest infiltration capacities were obtained for pavements with small proportions of joints and high degree of surface clogging. The methods used to derive soil hydrological parameters proved their ability, as all parameters lie within the range reported in literature. Recorded soil temperatures revealed that temperature variations underneath permeable pavements are more pronounced than underneath vegetated surfaces. Thereby, differences are most pronounced during the summer, when pavements undergo a stronger heating during daytime. Our measurements further indicate that the penetration depth of temperature variations is higher on paved plots compared to vegetated plots. Besides the surface type, the surrounding urban structures are a main control of soil temperatures. While soil temperatures reached 47°C on a paved plot exposed to direct solar irradiation, recorded maximum temperatures were 17°C lower at a shaded pavement. Furthermore, surface shading reduces the temperature difference between vegetated and paved plots. While the difference is pronounced at clusters exposed to direct solar irradiation, it is marginal at shaded clusters.

The provided dataset and the parameters derived within this study are valuable for various purposes. One possible application is the usage for calibration and validation of urban hydrological and climatological models. Further applications include the derivation of water and energy fluxes. Hence, the dataset enables to study the role of permeable pavements within urban storm water management practices. Furthermore, we encourage its usage to analyze the ground heat flux of urban areas. This might be of special interest for studying the urban energy balance but also for analyzing urban subsurface heat islands. Usage of the data is not only limited to the fields of urban hydrology and climatology, but also may be of interest for urban planning and geoengineering. Against this background, we are convinced that the provided dataset is of great value for various disciplines and scientific issues.

## Appendix A

**Table 5: Fraction [%] of different urban structures in the surrounding of each cluster**

|   | 5 m radius | | | | 10 m radius | | | |
|---|---|---|---|---|---|---|---|---|
|   | Buildings | PPs | Asphalt | Green spaces | Buildings | PPs | Asphalt | Green spaces |
| A | 7 | 56 | 0 | 36 | 11 | 46 | 0 | 43 |
| B | 0 | 66 | 0 | 34 | 12 | 58 | 9 | 21 |
| C | 0 | 58 | 0 | 42 | 5 | 40 | 0 | 55 |
| D | 34 | 66 | 0 | 0 | 42 | 58 | 0 | 0 |
| E | 19 | 51 | 30 | 0 | 31 | 44 | 25 | 0 |
| F | 11 | 63 | 25 | 0 | 26 | 50 | 24 | 0 |
| G | 0 | 56 | 44 | 0 | 0 | 42 | 41 | 17 |
| H | 1 | 72 | 0 | 27 | 23 | 56 | 0 | 20 |

**Table 6: Mean soil moisture [vol.%] measured at different plot categories and depths together with the mean standard deviation of each group**

|   | Depth below ground surface | | |
|---|---|---|---|
|   | <15 cm | 15-25 cm | >25 cm |
| **Vegetated** | 21.94 ± 4.95 | 22.04 ± 9.21 | 24.06 ± 4.29 |
| **PPs with free drainage** | 12.67 ± 3.04 | 11.93 ± 3.2 | 12.85 ± 5.72 |
| **PPs with restricted drainage** | 16.16 ± 2.64 | 19.2 ± 3.76 | 23.36 ± 4.04 |

**Table 7: Mean soil temperature [°C] measured at different plot categories and depths together with the mean standard deviation of each group**

|   | Depth below ground surface | | |
|---|---|---|---|
|   | <15 cm | 15-25 cm | >25 cm |
| **Vegetated** | 12.82 ± 1.08 | 12.92 ± 0.99 | 13.04 ± 1.01 |
| **PPs** | 14.66 ± 1.44 | 14.69 ± 1.29 | 14.44 ± 1.26 |

## Authors contribution

AS designed and maintained the sensor network and prepared the manuscript with contributions of all co-authors.

**Acknowledgements**

This study is part of the research project WaSiG (Wasserhaushalt siedlungsgeprägter Gewässer) which is part of joint project ReWaM (Regionales Wasserressourcen-Management für den nachhaltigen Gewässerschutz in Deutschland) funded by the German Ministry of Education and Research (BMBF). Furthermore, this work was supported by the badenova AG and Co.

KG (innovation fund for the protection of climate and water). Besides, we are greatly indebted to the local civil construction authority (Garten- und Tiefbauamt Freiburg i.Br.), who supported us during the installation of the soil moisture sensors. Furthermore, we want to thank the Center for Agricultural Technology Augustenberg and the State Viniculture Institute Freiburg for providing data for the WBI climate station. Finally, we like to express our gratitude to Caroline Siebert for performing most of the infiltration experiments.

**Literature**

Allen, R. G., Pereira, L. S., Raes, D. and Smith, M.: Crop evapotranspiration - Guidelines for computing crop water requirements, Rome, Italy., 1998.

Andersen, C. T., Foster, I. D. L. and Pratt, C. J.: The role of urban surfaces (permeable pavements) in regulating drainage and

evaporation: Development of a laboratory simulation experiment, Hydrol. Process., 13(4), 597–609, doi:10.1002/(SICI)1099-1085(199903)13:4<597::AID-HYP756>3.0.CO;2-Q, 1999.

Bogena, H. R., Huisman, J. A., Schilling, B., Weuthen, A. and Vereecken, H.: Effective calibration of low-cost soil water content sensors, Sensors (Switzerland), 17(1), doi:10.3390/s17010208, 2017.

Boogaard, F., Lucke, T., van de Giesen, N. and van de Ven, F.: Evaluating the infiltration performance of eight dutch permeable

pavements using a new full-scale infiltration testing method, Water (Switzerland), 6(7), 2070–2083, doi:10.3390/w6072070, 2014.

Borgwardt, S.: Merkblatt für wasserdurchlässige Befestigungen von Verkehrsflächen (der Forschungsgesellschaft für Straßen- und Verkehrswesen - FGSV) Kommentierung mit ausführlichen Hinweisen für die Planung und Ausführung versickerungsfähiger Bauweisen mit Betonpflaster, Fachvereinigung Betonprodukte für Straßen-, Landschafts- und

Gartenbau., 2001.

Borgwardt, S.: Long-Term In-Situ Infiltration Performance of Permeable Concrete Block Pavement, 8th Int. Conf. Concr. Block Paving, Novemb. 6-8, 2006 San Fr. Calif. USAnternational Conf. Concr. block paving, 149–160 [online] Available from: http://citeseerx.ist.psu.edu/viewdoc/download?doi=10.1.1.365.9174&rep=rep1&type=pdf, 2006.

Brocca, L., Melone, F. and Moramarco, T.: On the estimation of antecedent wetness conditions in rainfall-runoff modelling,

Hydrol. Process., 22(5), 629–642, doi:10.1002/hyp.6629, 2008.

Brown, R. A. and Borst, M.: Quantifying evaporation in a permeable pavement system, Hydrol. Process., 29(9), 2100–2111, doi:10.1002/hyp.10359, 2015.

Brunetti, G., Šimůnek, J. and Piro, P.: A comprehensive numerical analysis of the hydraulic behavior of a permeable pavement, J. Hydrol., 540, 1146–1161, doi:10.1016/j.jhydrol.2016.07.030, 2016.

Cristiano, E., Veldhuis, M. C. Ten and Van De Giesen, N.: Spatial and temporal variability of rainfall and their effects on hydrological response in urban areas - A review, Hydrol. Earth Syst. Sci., 21(7), 3859–3878, doi:10.5194/hess-21-3859-2017, 2017.

Demand, D., Selker, J. S. and Weiler, M.: Influences of Macropores on Infiltration into Seasonally Frozen Soil, , (1996), doi:10.2136/vzj2018.08.0147, 2019.

Eagleson, P. S.: Climate, soil, and vegetation: 3. A simplified model of soil moisture movement in the liquid phase, Water Resour. Res., 14(5), 722–730, doi:10.1029/WR014i005p00722, 1978.

Ekblad, J. and Isacsson, U.: Time-domain reflectometry measurements and soil-water characteristic curves of coarse granular materials used in road pavements, Can. Geotech. J., 44(7), 858–872, doi:10.1139/t07-024, 2007.

Elliott, A. H. and Trowsdale, S. A.: A review of models for low impact urban stormwater drainage, Environ. Model. Softw., 22(3), 394–405, doi:10.1016/j.envsoft.2005.12.005, 2007.

Fassman, E. A. and Blackbourn, S.: Urban Runoff Mitigation by a Permeable Pavement System over Impermeable Soils, J. Hydrol. Eng., 15(6), 475–485, doi:10.1061/(ASCE)HE.1943-5584.0000238, 2010.

FGSV: Richtlinien für die Standardisierung des Oberbaus., 2012.

Fletcher, T. D., Andrieu, H. and Hamel, P.: Understanding, management and modelling of urban hydrology and its consequences for receiving waters: A state of the art, Adv. Water Resour., 51, 261–279, doi:10.1016/j.advwatres.2012.09.001, 2013.

Grimmond, C. S. B. and Oke, T. R.: An evapotranspiration-interception model for urban areas, Water Resour. Res., 27(7), 1739–1755, doi:10.1029/91WR00557, 1991.

Grimmond, C. S. B., Blackett, M., Best, M. J., Barlow, J., Baik, J. J., Belcher, S. E., Bohnenstengel, S. I., Calmet, I., Chen, F., Dandou, A., Fortuniak, K., Gouvea, M. L., Hamdi, R., Hendry, M., Kawai, T., Kawamoto, Y., Kondo, H., Krayenhoff, E. S., Lee, S. H., Loridan, T., Martilli, A., Masson, V., Miao, S., Oleson, K., Pigeon, G., Porson, A., Ryu, Y. H., Salamanca, F., Shashua-Bar, L., Steeneveld, G. J., Tombrou, M., Voogt, J., Young, D. and Zhang, N.: The international urban energy balance models comparison project: First results from phase 1, J. Appl. Meteorol. Climatol., 49(6), 1268–1292, doi:10.1175/2010JAMC2354.1, 2010.

Guo, R., Guo, Y. and Wang, J.: Stormwater capture and antecedent moisture characteristics of permeable pavements, Hydrol. Process., 32(17), 2708–2720, doi:10.1002/hyp.13213, 2018.

Hillel, D.: Environmental Soil Physics, Acad. Press, San Diego, Calif., 1998.

Illgen, M.: Das Versickerungsverhalten durchlässig befestigter Siedlungsflächen und seine urbanhydrologische Quantifizierung., 2009.

Illgen, M., Harting, K., Schmitt, T. G. and Welker, A.: Runoff and infiltration characteristics of pavement structures-review of an extensive monitoring program, Water Sci. Technol., 56(10), 133–140, doi:10.2166/wst.2007.750, 2007.

Kapilaratne, R. G. C. J. and Lu, M.: Automated general temperature correction method for dielectric soil moisture sensors, J. Hydrol., 551, 203–216, doi:10.1016/j.jhydrol.2017.05.050, 2017.

Kimball, B. A., Jackson, R. D., Nakayama, F. S., Idso, S. B. and Reginato, R. J.: Soil-heat flux determination: temperature gradient method with computed thermal conductivities, Soil Sci. Soc. Am. J., 40(1), 25–28, doi:10.2136/sssaj1976.03615995004000010011x, 1976.

Kodešová, R., Fér, M., Klement, A., Nikodem, A., Teplá, D., Neuberger, P. and Bureš, P.: Impact of various surface covers on water and thermal regime of Technosol, J. Hydrol., 519(PB), 2272–2288, doi:10.1016/j.jhydrol.2014.10.035, 2014.

Lahoz, W. A. and De Lannoy, G. J. M.: Closing the Gaps in Our Knowledge of the Hydrological Cycle over Land: Conceptual Problems., 2014.

Litvak, E., Manago, K. F., Hogue, T. S. and Pataki, D. E.: Evapotranspiration of urban landscapes in Los Angeles, California at the municipal scale, Water Resour. Res., 53(5), 4236–4252, doi:10.1002/2016WR020254, 2017.

LUBW: Wasser- und Bodenatlas Baden-Württemberg (WaBoA), Ministry of Environment Baden-Württemberg., 2012.

Lucke, T. and Beecham, S.: Field investigation of clogging in a permeable pavement system, Build. Res. Inf., 39(6), 603–615, doi:10.1080/09613218.2011.602182, 2011.

Matzarakis, A., Rutz, F., Matzarakis, A., Rutz, F. and Mayer, H.: Modelling radiation fluxes in simple and complex environments - Application of the RayMan model Modelling radiation fluxes in simple and complex, Int J Biometeorol, 51, 323–334, doi:10.1007/s00484-006-0061-8, 2007.

Menberg, K., Blum, P., Schaffitel, A. and Bayer, P.: Long-term evolution of anthropogenic heat fluxes into a subsurface urban heat island, Environ. Sci. Technol., 47(17), 9747–9755, doi:10.1021/es401546u, 2013.

Oke, T. R.: The urban energy balance, Prog. Phys. Geogr., 12(4), 471–508, doi:10.1177/030913338801200401, 1988.

Oke, T. R., Mills, G., Christen, A. and Voogt, J. A.: Urban Climates, Cambridge University Press, Cambridge., 2017.

Or, D. and Wraith, J. M.: Temperature effects on soil bulk dielectric permittivity measured by time domain reflectometry: A physical model, Water Resour. Res., 35(2), 371–383, doi:10.1029/1998WR900008, 1999.

Park, D. G., Sandoval, N., Lin, W., Kim, H. and Cho, Y. H.: A case study: Evaluation of water storage capacity in permeable 20    block pavement, KSCE J. Civ. Eng., 18(2), 514–520, doi:10.1007/s12205-014-0036-y, 2014.

Philip, J. R.: The theory of infiltration: 4. Sorptivity and algebraic infiltration equations, Soil Sci., 84(3), 257–264, 1957.

Qu, W., Bogena, H. R., Huisman, J. A. and Vereecken, H.: Calibration of a Novel Low-Cost Soil Water Content Sensor Based on a Ring Oscillator, Vadose Zo. J., 12(2), 0, doi:10.2136/vzj2012.0139, 2013.

Ragab, R., Bromley, J., Rosier, P., Cooper, J. D. and Gash, J. H. C.: Experimental study of water fluxes in a residential area: 25    1. Rainfall, roof runoff and evaporation: the effect of slope and aspect, Hydrol. Process., 17(12), 2409–2422, doi:10.1002/hyp.1250, 2003.

Razzaghmanesh, M. and Borst, M.: Investigation clogging dynamic of permeable pavement systems using embedded sensors, J. Hydrol., 557, 887–896, doi:10.1016/j.jhydrol.2018.01.012, 2018.

Roberts, S. M., Oke, T. R., Grimmond, C. S. B. and Voogt, J. A.: Comparison of four methods to estimate urban heat storage, 30    J. Appl. Meteorol. Climatol., 45(12), 1766–1781, doi:10.1175/JAM2432.1, 2006.

Roth, K. and Boike, J.: Quantifying the thermal dynamics of a permafrost site near Ny-Ålesund, Svalbard, Water Resour. Res., 37(12), 2901–2914, doi:10.1029/2000WR000163, 2001.

Roth, K., Schulin, R., Flühler, H. and Attinger, W.: Calibration of time domain reflectometry for water content measurement using a composite dielectric approach, Water Resour. Res., 26(10), 2267–2273, doi:10.1029/WR026i010p02267, 1990.

Salvadore, E., Bronders, J. and Batelaan, O.: Hydrological modelling of urbanized catchments: A review and future directions,

J. Hydrol., 529(P1), 62–81, doi:10.1016/j.jhydrol.2015.06.028, 2015.

Schaffitel, A., Schuetz, T. and Weiler, M.: A distributed soil moisture, temperature and infiltrometer dataset for permeable pavements and green spaces, FreiDok plus; available at: https://doi.org/10.6094/UNIFR/157573, https://freidok.uni-freiburg.de/data/151573, doi:10.6094/UNIFR/151573, 2019.

Schirmer, M., Leschik, S. and Musolff, A.: Current research in urban hydrogeology - A review, Adv. Water Resour., 51, 280–291, doi:10.1016/j.advwatres.2012.06.015, 2013.

Scholz, M. and Grabowiecki, P.: Review of permeable pavement systems, Build. Environ., 42(11), 3830–3836, doi:10.1016/j.buildenv.2006.11.016, 2007.

Shuster, W. D., Bonta, J., Thurston, H., Warnemuende, E. and Smith, D. R.: Impacts of impervious surface on watershed
hydrology: A review, Urban Water J., 2(4), 263–275, doi:10.1080/15730620500386529, 2005.

Stander, E. K., Rowe, A. A., Borst, M. and O'Connor, T. P.: Novel Use of Time Domain Reflectometry in Infiltration-Based Low Impact Development Practices, J. Irrig. Drain. Eng., 139(8), 625–634, doi:10.1061/(ASCE)IR.1943-4774.0000595, 2013.

Timm, A., Kluge, B. and Wessolek, G.: Hydrological balance of paved surfaces in moist mid-latitude climate – A review, Landsc. Urban Plan., 175(April 2017), 80–91, doi:10.1016/j.landurbplan.2018.03.014, 2018.

Topp, G. C., Davis, J. L. and Annan, A. P.: Electromagnetic determination of soil water content: Measurements in coaxial transmission lines, Water Resour. Res., 16(3), 574–582, doi:10.1029/WR016i003p00574, 1980.

Trenberth, K. E. and Asrar, G. R.: Challenges and Opportunities in Water Cycle Research: WCRP Contributions, Surv. Geophys., 35(3), 515–532, doi:10.1007/s10712-012-9214-y, 2014.

Turco, M., Kodešová, R., Brunetti, G., Nikodem, A., Fér, M. and Piro, P.: Unsaturated hydraulic behaviour of a permeable
pavement: Laboratory investigation and numerical analysis by using the HYDRUS-2D model, J. Hydrol., 554, 780–791, doi:10.1016/j.jhydrol.2017.10.005, 2017.

Vereecken, H., Huisman, J. A., Hendricks Franssen, H. J., Brüggemann, N., Bogena, H. R., Kollet, S., Javaux, M., Van Der Kruk, J. and Vanderborght, J.: Soil hydrology: Recent methodological advances, challenges, and perspectives, Water Resour. Res., 51(4), 2616–2633, doi:10.1002/2014WR016852, 2015.

Weiler, M. and Flühler, H.: Inferring flow types from dye patterns in macroporous soils, Geoderma, 120(1–2), 137–153, doi:10.1016/j.geoderma.2003.08.014, 2004.

Wessolek, G.: Sealing of soils, in Urban Ecology: An International Perspective on the Interaction Between Humans and Nature, pp. 161–179, Springer US, Boston, MA., 2008.

Winston, R. J., Al-Rubaei, A. M., Blecken, G. T., Viklander, M. and Hunt, W. F.: Maintenance measures for preservation and
recovery of permeable pavement surface infiltration rate - The effects of street sweeping, vacuum cleaning, high pressure washing, and milling, J. Environ. Manage., 169, 132–144, doi:10.1016/j.jenvman.2015.12.026, 2016.

Wraith, J. M. and Or, D.: Temperature effects on soil bulk dielectric permittivity measured by time domain reflectometry: Experimental evidence and hypothesis development, Water Resour. Res., 35(2), 361–369, doi:10.1029/1998WR900006, 1999.

Zhu, K., Blum, P., Ferguson, G., Balke, K. D. and Bayer, P.: The geothermal potential of urban heat islands, Environ. Res.
Lett., 5(4), doi:10.1088/1748-9326/5/4/044002, 2010.