# Peer review of "A distributed soil moisture, temperature and infiltrometer dataset for permeable pavements and green spaces"

_Earth System Science Data, 2019_

## Referee Comment (RC1) · Heye Bogena (Referee) · 27 Jun 2019

This manuscript presents an interesting dataset on soil moisture, temperature and infiltrometer for permeable pavements and green spaces for the analysis urban hydrology in the city of Freiburg. The article is also very well written, and gives a very good overview of the presented data. The tables and figures are very informative and the data is archived in an appropriate way.

This is a quite unique data set for studying urban water and energy cycles and will be useful for the parameterization and testing of urban hydrological models. I have only a few general comments that should be considered.

[Figure]

General comments

Some of the sensors were installed within an excavated hole which then was refilled successively with bedding material. What kind of material did you use? If it is different from the site material in terms of soil hydraulic properties this could have led to biased measurements.

You applied the CRIM model by also considering the temperature dependency of permittivity. The same procedure was applied to the SMT100 sensor by Bogena et al. (2017) and they found that the derived soil moisture from the permittivity measured by the SMT100 did not show temperature effects. This indicated that the temperature effect was only due to the temperature dependence the permittivity and that the sensor electronics were not affected by temperature. Please discuss reasons for the remaining diurnal soil water content oscillations.

You removed data from frozen soils with the argument that freezing hinders vertical water movement within the profile. However the main reason should be that the dielectric properties of frozen water are different from the liquid water for which reason soil water content measurements with electromagnetic sensor of frozen soils are not reliable.

Finally, some remarks on the transferability of the data to other urban areas would be helpful for potential users of the data.

---

## Referee Comment (RC2) · Anonymous Referee #2 · 5 Sep 2019

Review ESSD-2019-97, urban surfaces hydrological properties

Overall: The authors have provided an interesting data product from carefully-conducted measurements in urban settings. I applaud their intention to share these data openly; I hope these data have impact and use as the authors list (e.g. p3l20). However, this potential user would need additional information and has some questions. Understanding that, at this point, the authors probably can not 'go back' to remedy some of these issues, this reviewer expects a more overt mention and discussion of uncertainties and weaknesses, partly to assist users of these data but also - to emphasize a good point made by the other reviewer - to guide future observations of similar properties in other urban settings.

The specific language seems quite awkward and potentially distracting in places. I itemize some of those errors below but I have no doubt that I missed many of them. These arise at least in part from German-to-English mis-translations. The journal / publisher will pick up some of these errors at the proof-reading step but I think that responsibility for these corrections lies with authors, not the journal. I strongly recommend that the authors engage a scientific technical editor to read and revise this text.

ESSD, according to it guidelines (https://www.earth-syst-sci-data.net/10/2275/2018/) requires explicit detailed description of uncertainty factors plus careful validation. I understand that, due to the unique nature and scale of these urban measurements, validation may prove difficult. However, the manuscript as presented remains woefully deficient on uncertainties.

These authors seem to assign uncertainty solely to sensor performance. For example, at page 10 lines 10 to 12, the authors merely recite manufacturer's performance data. But in fact they have a whole cascade of uncertainties among which manufacturer sensor performance may prove small. The climate source data (from WBI) must have substantial uncertainties. At a quick glance one sees many RH values near or at 100%, values in the highly-uncertain range for most humidity sensors.

(One often finds in this text file as well as in several others, very strange formatting errors, e.g air temperatures of 4.0489999999999995 or, in the metaPlot.txt file, GPS values of 7.8509169190999994. These easy-to-fix errors, not fixed in these cases, have the effect of eroding our confidence in the data generally. If the authors failed to find and fix these errors, what else have they missed?)

In addition to sensor imprecisions, one needs to add uncertainties in the source climate data, variability in specific locations and PP types, uncertainties in the infiltration measurements, uncertainties in application of the CRIM equation, etc. Some of these might cancel or offset, but a rigorous uncertainty analysis necessitates careful accounting of the full range of uncertainty factors. I do not contend that users should consider any of these data as 'wrong' but neither should we consider them - as these authors apparently do - as absolute. Soil moisture, soil temperature, saturated water content, etc. all have associated uncertainties. Readers need to know those uncertainties, need to know that the data providers recognize those uncertainties, and need to know - as we currently can not - how large an impact those uncertainties might or might not have on the validity of these data. Figures 6 through 9, which ought to help us understand the value of these data, have no indications of uncertainty. Files of permititivity, soil moisture, soil temperatue, etc., have no indications of uncertainty. Several times the authors mention "means" of all locations or all depths, but we never read nor see anything about standard deviations, standard errors, etc. The authors hope to see these data useful in the context of model calibration or validation, but most models require quantified uncertainty ranges. At the top of page 19 (lines 2 thru 4), the authors write "The plots E1 and E2 are equal in terms of joint properties and proportions, which leads to the assumption that infiltration measured at E1 might be applied as representative for E2." I appreciate that the

authors used the cautionary word 'might' but this reader find no basis elsewhere in the text, particularly assurances on uncertainties, that would allow me to accept similarity of E1 and E2.

A large uncertainty factor, at least for this user/reviewer, relates to solar exposure. How much direct solar radiation or shading by buildings or vegetation occurred at any site? For these latitudes, shade can influence soil temperatures by 10C or more, e.g. 50% or more of total diurnal ranges described here. Intensity of shade, diurnal pattern of shade, seasonal pattern of shade - we get none of this information and - apparently - no hints about how we might retrieve such data. Clearly the authors know more about solar radiation and local exposure factors than any users will ever know, but we get nothing?

On page 17 line 12 one reads about station D as located "an east-west orientated urban canyon within the city center." Using lat lon coordinates from metaPlots.txt file to locate the stations in Google Earth, and then applying the GE 'street view' function, I confirm the narrow streets and tall-ish buildings around station D, but I also find more dispersed but taller (5 or 6 stories?) buildings around station H, albeit with different E-W N-S orientations. From those two explorations (which I might have done wrongly, see note about lat lon below), this reader remains just as concerned and perhaps more concerned about insolation and shading effects. Authors must have recognized insolation effects, must have assessed and selected locations with solar exposure in mind, but they have shared none of that information with readers? They offer readers neither tools nor information needed to assess such a large uncertainty factor?

Again note errors in metaPlots.txt file: latitude and longitude apparently erroneously reversed for stations G and H. If authors did not recognize nor correct such an obvious error, what else have they missed? Another warning to users: data quality not assured?

Although this review seems harsh, I believe the authors have provided a potentially valuable data set that deserves publication and exposure through ESSD. To merit that publication, however, the authors need to provide a much better, more-thorough and confidence-building assessment of uncertainties.

Text errors:

Page 2 line 9: "alternated" should be 'altered'.
Page 6 line 7: "Thereby" ???
Page 7 line 2: "(see chapter data availability", 'chapter' as used here refers to a book or thesis, not to this paper?
Page 7 line 14: Authors mention evapotranspiration here (and provide two data files, one daily and one hourly) but then make no further mention or use of the term or the data. Again, a residual remaining from a separate publication or thesis?
Page 11 lines 6 thru 12: Here the authors describe uncertainties related to freezing conditions and possible salt applied as anti-freeze, e.g. reasons for not using winter-time data, but we never find any cautions about uses of the data they do provide!
Page 11 line 20: "flashy" ???
Page 11 lines 25, 26: characterization as vegetated, restricted or free. But, according to Table 2, they only analyzed 3 vegetated sites and 4 restricted drainage sites. Given many other sources of spatial variability and uncertainty, can the authors provide any quantitative basis that we should accept these categorizations?

Etc. Frankly, this review got tired of identifying and commented on all these errors. I repeat a recommendation from above: "engage a scientific technical editor to read and revise this text".

---

## Author Comment (AC1) · 31 Oct 2019

**Response to the comments of referee#1 (Heye Bogena) on the manuscript "A distributed soil moisture, temperature and infiltrometer dataset for permeable pavements and green spaces"**

We thank Heye Bogena for reviewing our manuscript, for his positive overall evaluation and for his helpful suggestions for improving the manuscript. In the following, we answer the comments in a point-by-point reply.

**R1 C1:** *Some of the sensors were installed within an excavated hole, which then was refilled successively with bedding material. What kind of material did you use? If it is different from the site material in terms of soil hydraulic properties this could have led to biased measurements.*

> Thank you for this point, which we will clarify in the manuscript. At the plots, where sensors were installed within the excavated hole, we used bedding material (material used for the construction of the bedding layer) to refill these holes. The refilling and compaction was done professionally by municipal construction workers. Within the bedding layer, soil hydraulic properties between refilling and original soil material should therefore be comparable. Soils found within the underlying base and subbase layers were characterized by a strong variability. We think that the hydraulic properties of the refilling lie within this variability.

**R1 C2:** *You applied the CRIM model by also considering the temperature dependency of permittivity. The same procedure was applied to the SMT100 sensor by Bogena et al. (2017) and they found that the derived soil moisture from the permittivity measured by the SMT100 did not show temperature effects. This indicated that the temperature effect was only due to the temperature dependence the permittivity and that the sensor electronics were not affected by temperature. Please discuss reasons for the remaining diurnal soil water content oscillations*

> Indeed, this is a very interesting issue. Diurnal soil water content oscillations present in our data are most pronounced at sensors near the ground surface and are characterized by an increase in soil moisture with rising soil temperature.
>
> Temperature effects electromagnetic soil moisture measurements in various ways. Wraith and Or (1999) showed that temperature effects may depend on soil type and soil moisture content. This is explained by the effect of bound water, which gets partially released when temperatures increase (Or and Wraith, 1999). While the effect of bound water may be negligible at high soil moisture contents, it might play an important role at low soil moisture contents. In our data, the diurnal soil moisture oscillations occur at low soil moisture contents. In contrast, the study of Bogena et al. (2017) took place at much

higher soil moisture contents and therefore the effect of bound water may be less present in their observations.

**R1 C3:** *You removed data from frozen soils with the argument that freezing hinders vertical water movement within the profile. However the main reason should be that the dielectric properties of frozen water are different from the liquid water for which reason soil water content measurements with electromagnetic sensor of frozen soils are not reliable.*

> Thank you for this remark, which we will consider in the manuscript. Note that although times with freezing conditions were removed in the soil moisture data, these times are still present in the permittivity data. Augmenting the CRIM model by the permittivity of frozen water (e.g. applied by Demand et al. (2019) and by Roth and Boike (2001)) enables to calculate the liquid water content even for frozen conditions. We will add a corresponding remark in the manuscript.

**R1 C4:** *Some remarks on the transferability of the data to other urban areas would be helpful for potential users of the data*

> We agree that such information will improve the manuscript. We will therefore add the following remarks to the manuscript:
>
> Urban areas are characterized by strong spatial heterogeneities concerning surface coverage but also in regard to urban soils. Thereby, the heterogeneity of the urban surface coverage leads to a variable input at the ground surface (e.g. insolation and precipitation). In combination with the pronounced variability of urban soil composition, this aggravates the transferability of the data to other urban sites.
>
> Nevertheless, the soil layers underneath PPs consist of technical substrates with defined hydrological properties. Therefore, we expect that soil moisture patterns underneath PPs are similar and that the observed patterns could be transferred to other field sides. This is also the case for the parameters derived from soil moisture measurements ($\theta_s$ and $\theta_{fc}$).
>
> In contrast, various authors highlighted the variability of the infiltration capacity of PPs (e.g. Illgen (2009)). Therefore, the transferability of the infiltrometer data is limited.

**Literature**

Bogena, H. R., Huisman, J. A., Schilling, B., Weuthen, A. and Vereecken, H.: Effective calibration of low-cost soil water content sensors, Sensors (Switzerland), 17(1), doi:10.3390/s17010208, 2017.

Demand, D., Selker, J. S. and Weiler, M.: Influences of Macropores on Infiltration into Seasonally Frozen Soil, , (1996), doi:10.2136/vzj2018.08.0147, 2019.

Illgen, M.: Das Versickerungsverhalten durchlässig befestigter Siedlungsflächen und seine urbanhydrologische Quantifizierung., 2009.

Or, D. and Wraith, J. M.: Temperature effects on soil bulk dielectric permittivity measured by time domain reflectometry: A physical model, Water Resour. Res., 35(2), 371–383, doi:10.1029/1998WR900008, 1999.

Roth, K. and Boike, J.: Quantifying the thermal dynamics of a permafrost site near Ny-Ålesund, Svalbard, Water Resour. Res., 37(12), 2901–2914, doi:10.1029/2000WR000163, 2001.

Wraith, J. M. and Or, D.: Temperature effects on soil bulk dielectric permittivity measured by time domain reflectometry: Experimental evidence and hypothesis development, Water Resour. Res., 35(2), 361–369, doi:10.1029/1998WR900006, 1999.

---

## Author Comment (AC2) · 31 Oct 2019

**Response to the comments of referee#2 (anonymous) on the manuscript "A distributed soil moisture, temperature and infiltrometer dataset for permeable pavements and green spaces"**

We thank referee#2 for the valuable comments that will help us in improving the quality and readably of the manuscript. We are deeply grateful for that. We assigned the comments into the three categories text errors, provided data and data uncertainty.

**Text errors**

*R2 C1: The specific language seems quite awkward and potentially distracting in places. I itemize some of those errors below but I have no doubt that I missed many of them. These arise at least in part from German-to-English mis-translations. The journal / publisher will pick up some of these errors at the proof-reading step but I think that responsibility for these corrections lies with authors, not the journal. I strongly recommend that the authors engage a scientific technical editor to read and revise this text.*

> We will revise the whole manuscript and have it checked by an English native speaker to improve its readability.

*P2, L9: "alternated" should be altered*

> Corrected

*P6, L7: "Thereby"*

> Changed into this

*P7, L2: "see chapter data availability", 'chapter' as used here refers to a book or thesis, not to this paper*

> According to the author guidelines of ESSD, the abbreviation Sect. is used throughout the manuscript instead of "chapter"

*P11, L20: flashy?*

> Explanation added in the manuscript (fast rise and recession of soil moisture)

**Provided data**

**R2 C2:** *Page 7 line 14: Authors mention evapotranspiration here (and provide two data files, one daily and one hourly) but then make no further mention or use of them or the data. Again, a residual remaining from a separate publication or thesis?*

In the following, we explain the reasons for providing reference crop evaporation ($et_0$) with different temporal resolutions. We will add these reasons in the manuscript.

Although not analyzed within the manuscript, we decided to provide $et_0$ since it is a key variable for most hydrological studies. Since a high temporal resolution might be desirable for further users, we provide $et_0$ with an hourly temporal resolution. However, the time step recommended by Allen et al. (1998) for the calculation of $et_0$ is one day. Therefore, we decided to include also daily values for $et_0$.

**R2 C3:** *One often finds in this text file as well as in several others, very strange formatting errors, e.g air temperatures of 4.0489999999999995 or, in the metaPlot.txt file, GPS values of 7.8509169190999994.*
*Note errors in metaPlots.txt file: latitude and longitude apparently erroneously reversed for stations G and H*

Thank you for pointing out these formatting errors, which will be corrected in the data files. The plot coordinates will be provided with 6 decimal digits, while e.g. air temperature will be provided with a precision of 2 digits.

**Uncertainties**

*R2 C4: ESSD, according to it guidelines (https://www.earth-syst-sci-data.net/10/2275/2018/) requires explicit detailed description of uncertainty factors plus careful validation. I understand that, due to the unique nature and scale of these urban measurements, validation may prove difficult. However, the manuscript as presented remains woefully deficient on uncertainties.*
*The authors seem to assign uncertainty solely to sensor performance. For example, at page 10 lines 10 to 12, the authors merely recite manufacturer's performance data. But in fact they have a whole cascade of uncertainties among which manufacturer sensor performance may prove small.*
*A rigorous uncertainty analysis necessitates careful accounting of the full range of uncertainty factors. I do not contend that users should consider any of these data as 'wrong' but neither should we consider them - as these authors apparently do - as absolute. Soil moisture, soil temperature, saturated water content, etc. all have associated uncertainties. Readers need to know those uncertainties, need to know that the data providers recognize those uncertainties, and need to know - as we currently can not - how large an impact those uncertainties might or might not have on the validity of these data.*
*The authors hope to see these data useful in the context of model calibration or validation, but most models require quantified uncertainty ranges.*

Preliminary notes:
Indeed, there are different sources of uncertainties affecting the measurements and the parameters presented within this manuscript. We are grateful for the comments of reviewer#2 highlighting these uncertainties. Since the data originate from point measurements, we focus on the uncertainties of the measured and derived quantities, while problems of scale and time are not discussed.

*R2 C5: The climate source data (from WBI) must have substantial uncertainties. At a quick glance one sees many RH values near or at 100%, values in the highly-uncertain range for most humidity sensors.*

There are four different climate stations available for the study area, which are all operated by different institutions. In the manuscript, we provide a link to the data of each station. The focus of the manuscript is on soil moisture, soil temperature and infiltrometer data. Hence, we think that a comprehensive discussion of climate data uncertainty is beyond the scope of this manuscript. However, we will include the following remarks into the manuscript:

Data of the individual climate stations differ in resolution, documentation, provided variables and vicinity to soil moisture clusters. Therefore, the selection of the climate data should be purpose-specific. Documentation is best for the DWD climate station, which is operated according to guidelines of the World Meteorological Organization. For the WBI climate station, online available data is limited to an hourly temporal resolution, while data with a higher temporal resolution is available only upon request. To facilitate the use of high resolution

data for the WBI climate station and to ensure its long-term availability, we asked for the permission to include this data in our data repository.

Note that the link to DWD data has changed and will be updated in the manuscript. Furthermore the station-ID will be added.

**R2 C6: *Need to add variability in specific locations and PP types**

Indeed, this variability may be important for interpreting the data. We will provide this information by:

- Adding images of the PP surfaces to the data repository which show the variability of the PP surface. These images cover an area of 1 m² and consist of digitized paving stones (black) and joints (white).

- Adding a file metaClusters.txt to the data repository. For each cluster, this file will contain a column with the fraction of different urban structures (buildings, asphalt, PPs and green spaces) within a 5 m and 10 m radius around the clusters. This data will be obtained by means of a GIS analysis and will capture the variability in urban structures in the surrounding of each cluster.

**R2 C7: *Need to add uncertainties in the infiltration measurements**

For the infiltration data, uncertainties comprise the measurement accuracy (approx. 0.5 mm for the visual observations) and the parameter uncertainties of the fitted Philip infiltration model. While measurement errors over the entire infiltration course are small (measurement of a cumulative quantity), the latter is included in table 3 of the manuscript and will be further included in the file metaPlots.txt. For sake of the clarity, we decided to not include uncertainty bands for the fitted Philip model in Figure 10.

**R2 C8: *Need to add uncertainties in the CRIM equation**

Given that the CRIM equation correctly describes the physics of the system, uncertainties arise from measurement errors (dielectric permittivity and temperature) and parameter uncertainties (porosity, permittivity of the gaseous and solid phase). By estimating these input uncertainties, Roth et al. (1990) calculated their effect to not exceed 1.3 vol.%. We will add this uncertainty in the manuscript.

*R2 C9: Figures 6 through 9, which ought to help us understand the value of the data, have no indications of uncertainty.*

Uncertainty bands will be added to Fig.6-9

*R2 C10: Files of permititivity, soil moisture, soil temperature, etc., have no indications of uncertainty.*

Indeed, the uncertainties should be indicated in the data files. We will add the uncertainty of each measured/derived variable in the comment line of the corresponding data file. Thereby, the error of the permittivity measurement, is estimated from the data reported by Bogena et al. (2017) who tested 701 SMT100 sensors in reference liquids with known dielectric properties. Their results indicate that the error of the permittivity measurement is below 1.5.

*R2 C11: Several times the authors mention "means" of all locations or all depths, but we never read nor see anything about standard deviations, standard errors, etc.*

Thank you for this comment. We will add the standard deviation for the median of the derived hydrologic parameters in Table 3 and in the text. Furthermore, Fig. 6-8 show mean values of soil moisture and soil temperatures over different sensors. Including the standard deviation in these Figures would be confusing. Instead, we will provide the standard deviation in a separate table.

*R2 C12: At the top of page 19 (lines 2 thru 4), the authors write "The plots E1 and E2 are equal in terms of joint properties and proportions, which leads to the assumption that infiltration measured at E1 might be applied as representative for E2." I appreciate that the authors used the cautionary word 'might' but this reader find no basis elsewhere in the text, particularly assurances on uncertainties, that would allow me to accept similarity of E1 and E2.*

We thank reviewer#2 for scrutinizing the assumed similarity in the infiltration patterns between plots E1/E2 and plots F1/F2. Due to the fact, that these plots were constructed during the same field campaigns (same age, similar soil material used for base and bedding layers), have the same proportion of joints and are exposed to similar microclimatological conditions, we think that this assumption is reasonable. However, since infiltration patterns may vary on small spatial scales, we will remove this assumption in the text and for the infiltration parameters (A, S and $i_{cap}$) in the file metaPlots.txt.

*R2 C13: A large uncertainty factor, at least for this user/reviewer, relates to solar exposure. How much direct solar radiation or shading by buildings or vegetation occurred at any site? For these latitudes, shade can influence soil temperatures by 10°C or more, e.g. 50% or more of total diurnal ranges described here. Intensity of shade, diurnal pattern of shade, seasonal pattern of shade - we get none of this information and - apparently - no hints about how we might retrieve such data. Clearly the authors know more about solar radiation and local exposure factors than any users will ever know, but we get nothing?*

*On page 17 line 12 one reads about station D as located "an east-west orientated urban canyon within the city center." Using lat lon coordinates from metaPlots.txt file to locate the stations in Google Earth, and then applying the GE 'street view' function, I confirm the narrow streets and tallish buildings around station D, but I also find more dispersed but taller (5 or 6 stories?) buildings around station H, albeit with different E-W N-S orientations. From those two explorations (which I might have done wrongly, see note about lat lon below), this reader remains just as concerned and perhaps more concerned about insolation and shading effects. Authors must have recognized insolation effects, must have assessed and selected locations with solar exposure in mind, but they have shared none of that information with readers? They offer readers neither tools nor information needed to assess such a large uncertainty factor?*

Indeed shading and insulation have a decisive effect on ground surface and subsurface temperatures. Adding the necessary information to the publication will certainly improve the usability of the data. Since shading and insulation is variable over time, the effect cannot be quantified by a single number. To enable for a time-dependent quantification of the effect, we will include hemispherical photos of each cluster in the data repository. From these images, the sun path, but also the mean radiant temperature can be calculated by using a suited model. One example for such a model is 'RayMan' (Matzarakis et al., 2007).

*C14: Page 11 lines 6 thru 12: Here the authors describe uncertainties related to freezing conditions and possible salt applied as anti-freeze, e.g. reasons for not using winter-time data, but we never find any cautions about uses of the data they do provide!*

The soil moisture data set does not contain data for freezing conditions (see P11 L6). However, the permittivity data set contains data for these periods. In the manuscript, we will emphasize that a suited model is required when relating permittivity to soil moisture for these periods (see answer to comment 3 of referee#1).

The other point concerns the usage of salt as an anti-freeze, which affects electromagnetic soil moisture measurements. Indeed, a further note on the uncertainty arising from this practice may be useful for other users and will be included in the manuscript.

***R2 C15:*** *P11, L25-26: Characterization as vegetated, restricted or free. But, according to Table 2, they only analyzed 3 vegetated sites and 4 restricted drainage sites. Given many other sources of spatial variability and uncertainty, can the authors provide any quantitative basis that we should accept these categorizations?*

Indeed, there is only a small number of plots for the categories "vegetated" and "PPs with restricted drainage". The small number of vegetated sites is due to the fact that only four of the clusters were in the close vicinity of urban green spaces.

The quantification into the categories free drainage and restricted drainage is based on a combination between a visual classification and an analysis of the empirical frequency distribution of soil moisture records during rain events. As illustrated in Fig. 7 the mode of these frequency distributions can be used as a quantitative measure for this classification. However, using a single threshold would neglect the effect of soil properties on soil moisture. In order to still provide some quantitative measure, we will add the mode of the empirical frequency distributions to the file metaPlots.txt.

**Literature**

Allen, R. G., Pereira, L. S., Raes, D. and Smith, M.: Crop evapotranspiration - Guidelines for computing crop water requirements, Rome, Italy., 1998.

Bogena, H. R., Huisman, J. A., Schilling, B., Weuthen, A. and Vereecken, H.: Effective calibration of low-cost soil water content sensors, Sensors (Switzerland), 17(1), doi:10.3390/s17010208, 2017.

Matzarakis, A., Rutz, F., Matzarakis, A., Rutz, F. and Mayer, H.: Modelling radiation fluxes in simple and complex environments - Application of the RayMan model Modelling radiation fluxes in simple and complex, Int J Biometeorol, 51, 323–334, doi:10.1007/s00484-006-0061-8, 2007.

Roth, K., Schulin, R., Flühler, H. and Attinger, W.: Calibration of time domain reflectometry for water content measurement using a composite dielectric approach, Water Resour. Res., 26(10), 2267–2273, doi:10.1029/WR026i010p02267, 1990.

---

## Author Response (AR1)

Dear editor,

Thank you for editing our manuscript. The first part of this document includes the point-by-point response to the reviews (R1, R2). Comments of the referees are marked as e.g. << R1 C1: *"referees' comment"*>> followed by the answer from the authors, which includes the changes made in the manuscript to fulfill the referees' suggestions.

The section of responses to the referees is followed by a marked-up version of the manuscript.

Best regards,
Schaffitel et al.

**Response to the comments of referee#1 (Heye Bogena) on the manuscript "A distributed soil moisture, temperature and infiltrometer dataset for permeable pavements and green spaces"**

We thank Heye Bogena for reviewing our manuscript, for his positive overall evaluation and for his helpful suggestions for improving the manuscript. In the following, we answer the comments in a point-by-point reply.

*R1 C1: Some of the sensors were installed within an excavated hole, which then was refilled successively with bedding material. What kind of material did you use? If it is different from the site material in terms of soil hydraulic properties this could have led to biased measurements.*

> Thank you for this comment. To clarify this point we added the following sentences to the manuscript: *"…soil hydraulic properties between the refilling and the original soil material should be comparable within the bedding layer. Soils found within the underlying layers were characterized by a strong variability and the hydraulic properties of the refilling should lie within the variability occurring within those layers."*

*R1 C2: You applied the CRIM model by also considering the temperature dependency of permittivity. The same procedure was applied to the SMT100 sensor by Bogena et al. (2017) and they found that the derived soil moisture from the permittivity measured by the SMT100 did not show temperature effects. This indicated that the temperature effect was only due to the temperature dependence the permittivity and that the sensor electronics were not affected by temperature. Please discuss reasons for the remaining diurnal soil water content oscillations*

> Indeed, this is a very interesting issue. To discuss this point, we added the following paragraph to the manuscript: *"After applying the CRIM model, temperature effects were still present in the $\theta$-time series. They manifest in form of diurnal soil moisture oscillations, which are characterized by increasing $\theta$ with rising $T_{soil}$. Their occurrence contrasts the findings of Bogena et al. (2017), who showed that $\theta$ derived by the CRIM model was free from temperature effects. A possible explanation for the different findings may be given by the effect of bound water, which gets partially released with increasing $T_{soil}$ (Or and Wraith, 1999). While the effect of bound water may be negligible at high $\theta$, it might play an important role at low $\theta$. In contrast to the study of Bogena et al. (2017), $\theta$ measured within this study reaches much lower values. In our dataset, temperature effects are most pronounced during those periods with low $\theta$, which supports the assumption that the observed temperature effects are caused by bound water."*

***R1 C3:*** *You removed data from frozen soils with the argument that freezing hinders vertical water movement within the profile. However the main reason should be that the dielectric properties of frozen water are different from the liquid water for which reason soil water content measurements with electromagnetic sensor of frozen soils are not reliable.*

Thank you for this remark, which we took into account as follows: "*Since dielectric properties of frozen and liquid water differ from that of liquid water, times with frozen soils were removed from the $\theta$-time series (but not from the $\varepsilon_c$ time series)... However, users interested in $\theta$ during those times may use an adapted version of the CRIM model, which enables to consider the dielectric properties of frozen water (examples for such an adapted version of the CRIM model are used e.g. by Demand et al. (2019) and by Roth and Boike (2001)).*

***R1 C4:*** *Some remarks on the transferability of the data to other urban areas would be helpful for potential users of the data*

We agree that such information will improve the manuscript. We therefore added the following paragraph to the manuscript: "*The presented dataset poses a valuable source of information for the urban environment. However, the pronounced heterogeneity in urban surface coverage and in urban soil composition aggravates the transferability of the dataset to other urban sites. However, since soil layers underneath PPs consist of technical substrates with defined hydrological properties, soil moisture patterns underneath PPs should be similar. Hence, the observed patterns should be transferable to other urban sites. This is also the case for the parameters derived from soil moisture measurements ($\theta_s$ and $\theta_{fc}$). In contrast, various authors highlighted the variability of the infiltration capacity of PPs, which decisively depends on the state of joint clogging (e.g. Illgen (2009)). Therefore, the transferability of the infiltrometer data is limited.* "

**Response to the comments of referee#2 (anonymous) on the manuscript "A distributed soil moisture, temperature and infiltrometer dataset for permeable pavements and green spaces"**

We thank referee#2 for the valuable comments that will help us in improving the quality and readably of the manuscript. We are deeply grateful for that. We assigned the comments into the three categories text errors, provided data and data uncertainty.

**Text errors**

*R2 C1: The specific language seems quite awkward and potentially distracting in places. I itemize some of those errors below but I have no doubt that I missed many of them. These arise at least in part from German-to-English mis-translations. The journal / publisher will pick up some of these errors at the proof-reading step but I think that responsibility for these corrections lies with authors, not the journal. I strongly recommend that the authors engage a scientific technical editor to read and revise this text.*

We revised the whole manuscript to improve its readability.

*P2, L9: "alternated" should be altered*

Corrected

*P6, L7: "Thereby"*

Changed into this

*P7, L2: "see chapter data availability", 'chapter' as used here refers to a book or thesis, not to this paper*

According to the author guidelines of ESSD, the abbreviation Sect. is used throughout the manuscript instead of "chapter"

*P11, L20: flashy?*

Explanation added in the manuscript (fast rise and recession of soil moisture)

**Provided data**

*R2 C2:* *Page 7 line 14: Authors mention evapotranspiration here (and provide two data files, one daily and one hourly) but then make no further mention or use of them or the data. Again, a residual remaining from a separate publication or thesis?*

We added the following explanations for providing reference crop evaporation ($et_0$) with different temporal resolutions in the manuscript:

*"Reference crop evapotranspiration ($et_0$) is a key variable for most hydrological studies and was calculated for the WBI climate station by using the Pennman-Monteith equation and the parametrization recommended by Allen et al. (1998). The time step recommended for the calculation of $et_0$ is one day (Allen et al., 1998). Since a high temporal resolution might be desirable for further users, we decided to provide $et_0$ also with an hourly temporal resolution".*

*R2 C3:* *One often finds in this text file as well as in several others, very strange formatting errors, e.g air temperatures of 4.0489999999999995 or, in the metaPlot.txt file, GPS values of 7.8509169190999994.*
*Note errors in metaPlots.txt file: latitude and longitude apparently erroneously reversed for stations G and H*

Thank you for pointing out these formatting errors, which we have corrected in the data files. Now, the plot coordinates are provided with 6 decimal digits, while e.g. air temperature is provided with a precision of 2 digits.

**Uncertainties**

*R2 C4:* *ESSD, according to it guidelines (https://www.earth-syst-sci-data.net/10/2275/2018/) requires explicit detailed description of uncertainty factors plus careful validation. I understand that, due to the unique nature and scale of these urban measurements, validation may prove difficult. However, the manuscript as presented remains woefully deficient on uncertainties.*
*The authors seem to assign uncertainty solely to sensor performance. For example, at page 10 lines 10 to 12, the authors merely recite manufacturer's performance data. But in fact they have a whole cascade of uncertainties among which manufacturer sensor performance may prove small.*
*A rigorous uncertainty analysis necessitates careful accounting of the full range of uncertainty factors. I do not contend that users should consider any of these data as 'wrong' but neither should we consider them - as these authors apparently do - as absolute. Soil moisture, soil temperature, saturated water content, etc. all have associated uncertainties. Readers need to know those uncertainties, need to know that the data providers recognize those uncertainties, and need to know - as we currently can not - how large an impact those uncertainties might or might not have on the validity of these data.*
*The authors hope to see these data useful in the context of model calibration or validation, but most models require quantified uncertainty ranges.*

Preliminary notes:
Indeed, there are different sources of uncertainties affecting the measurements and the parameters presented within this manuscript. We are grateful for the comments of reviewer#2 highlighting these uncertainties. Since the data originate from point measurements, we focus on the uncertainties of the measured and derived quantities, while problems of scale and time are not discussed.

*R2 C5:* *The climate source data (from WBI) must have substantial uncertainties. At a quick glance one sees many RH values near or at 100%, values in the highly-uncertain range for most humidity sensors.*

There are four different climate stations available for the study area, which are all operated by different institutions. In the manuscript, we provide a link to the data of each station. The focus of the manuscript is on soil moisture, soil temperature and infiltrometer data. Hence, we think that a comprehensive discussion of climate data uncertainty is beyond the scope of this manuscript. However, we included the following remarks in the manuscript:

*"Data of the individual climate stations differ in resolution, documentation, provided variables and vicinity to soil moisture clusters. Therefore, data users should select the climate data in dependence of their specific purpose… Since the DWD climate station is operated according to the guidelines of the World Meteorological Organization, the available documentation is best for this station and the measured climate variables should be unbiased by urban effects… In*

*order to facilitate the use of high resolution climate data for the WBI climate station and to ensure its long-term availability, we asked for the permission to include this data in our data repository."*

Note that the link to DWD data has changed and was updated in the manuscript. Furthermore, the station-ID was added.

**R2 C6:** *Need to add variability in specific locations and PP types*

Indeed, this variability may be important for interpreting the data. We provide this information by:

Adding images of the PP surfaces to the data repository which show the variability of the PP surface. These images cover an area of 1 m² and consist of digitized paving stones (black) and joints (white).We added the following remark to the manuscript: *"These digitized images capture the small scale variability of the PP surfaces and are included within the data repository."*

Adding a file metaClusters.txt to the data repository. For each cluster, this file contains a column with the fraction of different urban structures (buildings, asphalt, PPs and green spaces) within a 5 m and 10 m radius around the clusters. This data was obtained by means of a GIS analysis and captures the variability in urban structures in the surrounding of each cluster. In the manuscript, we added the following remark: *"…we analyzed the fraction of different urban structures within a 5 m and 10 m radius around each cluster by means of a GIS analysis. The results of this analysis are shown in Appendix A (Table 5) and are further included within the data repository (file metaClusters.txt)."*

**R2 C7:** *Need to add uncertainties in the infiltration measurements*

For the infiltration data, uncertainties comprise the measurement accuracy (approx. 0.5 mm for the visual observations) and the parameter uncertainties of the fitted Philip infiltration model. For the measurement accuracy, we added the following remark in the manuscript: *"The reading accuracy of the visual observations is approx. 0.5 mm. However, measurement errors should mainly cancel out over the infiltration course, since a cumulative quantity was measured."*

Uncertainties in the parameters of the fitted Philip model are found in table 3 of the manuscript and in the file metaPlots.txt. For sake of the clarity, we decided to not include uncertainty bands in Figure 11.

*R2 C8: Need to add uncertainties in the CRIM equation*

We added the uncertainty arising from the CRIM equation by including the following sentence in the manuscript: *"The effect of measurement errors (dielectric permittivity and temperature) and parameter uncertainties (porosity, permittivity of the gaseous and solid phase) on $\theta$ calculated by the CRIM equation was quantified by Roth et al. (1990) to not exceed 1.3 vol.%."*

*R2 C9: Figures 6 through 9, which ought to help us understand the value of the data, have no indications of uncertainty.*

Uncertainty is estimated to account ±0.4°C for soil temperature (based on specifications of the manufacturer) and ±1.3 vol.% for soil moisture calculated by the CRIM model (see R2, C8). Those uncertainties are included in the manuscript. Furthermore, we added comment line in each of the corresponding data files, which indicates the uncertainty.

Adding those uncertainties in Fig.6-9, would lead to a reduced readability of those Figures, while the gain in information is marginally. For sake of clarity, we therefore decided to not include uncertainty bands in Fig.6-9

*R2 C10: Files of permititivity, soil moisture, soil temperature, etc., have no indications of uncertainty.*

We added the uncertainty of each measured/derived variable in the comment line of the corresponding data file. For soil moisture, the uncertainty is given by the uncertainty of the CRIM model (see R2 C8), while for temperature measurements, the uncertainty is specified by the manufacturer to range between 0.2°C and 0.4°C. For the uncertainty in permittivity, we used the results of Bogena et al. (2017) and added the following sentence to the manuscript: *"…the error of the permittivity measurement is estimated from the results of Bogena et al. (2017) who tested 701 SMT100 sensors in reference liquids with known dielectric properties. Their results indicate that the error of the permittivity measurement is below 1.5."*

*R2 C11: Several times the authors mention "means" of all locations or all depths, but we never read nor see anything about standard deviations, standard errors, etc.*

Thank you for this comment. Figures 6-8 show mean values of soil moisture and soil temperature for different plot categories and depths. Including the standard deviation in these Figures would be confusing. In the Appendix (Table 6 & 7, we added tables which show the mean standard deviation between all sensors within each category.

Furthermore, we added the standard deviation for the median of the derived hydrologic parameters in Table 3.

*R2 C12: At the top of page 19 (lines 2 thru 4), the authors write "The plots E1 and E2 are equal in terms of joint properties and proportions, which leads to the assumption that infiltration measured at E1 might be applied as representative for E2." I appreciate that the authors used the cautionary word 'might' but this reader find no basis elsewhere in the text, particularly assurances on uncertainties, that would allow me to accept similarity of E1 and E2.*

We thank reviewer#2 for scrutinizing the assumed similarity in the infiltration patterns between plots E1/E2 and plots F1/F2. Due to the fact, that these plots were constructed during the same field campaigns (same age, similar soil material used for base and bedding layers), have the same proportion of joints and are exposed to similar microclimatological conditions, we think that this assumption is reasonable. However, since infiltration patterns may vary on small spatial scales, we removed this assumption in the text and further removed the infiltration parameters (A, S and $i_{cap}$) for the Plots E2 and F2 in table 3 and in the file metaPlots.txt.

*R2 C13: A large uncertainty factor, at least for this user/reviewer, relates to solar exposure. How much direct solar radiation or shading by buildings or vegetation occurred at any site? For these latitudes, shade can influence soil temperatures by 10°C or more, e.g. 50% or more of total diurnal ranges described here. Intensity of shade, diurnal pattern of shade, seasonal pattern of shade - we get none of this information and - apparently - no hints about how we might retrieve such data. Clearly the authors know more about solar radiation and local exposure factors than any users will ever know, but we get nothing?*
*On page 17 line 12 one reads about station D as located "an east-west orientated urban canyon within the city center." Using lat lon coordinates from metaPlots.txt file to locate the stations in Google Earth, and then applying the GE 'street view' function, I confirm the narrow streets and tallish buildings around station D, but I also find more dispersed but taller (5 or 6 stories?) buildings around station H, albeit with different E-W N-S orientations. From those two explorations (which I might have done wrongly, see note about lat lon below), this reader remains just as concerned and perhaps more concerned about insolation and shading effects. Authors must have recognized insolation effects, must have assessed and selected locations with solar exposure in mind, but they have shared none of that information with readers? They offer readers neither tools nor information needed to assess such a large uncertainty factor?*

Indeed shading and insolation have a decisive effect on ground surface and subsurface temperatures. Since shading and insolation is variable over time, the effect cannot be quantified by a single number. To enable for a time-dependent quantification of the effect, we included hemispherical photos of each cluster in the data repository. From these images, the sun path, but also the mean radiant temperature can be calculated by using a suited model. One example for such a model is 'RayMan' (Matzarakis et al., 2007). In the manuscript, we added the following remark:

*"Hemispherical photos were taken at each cluster and are included within the data repository. Using those photos allows to calculate the sun path for each day of the year (see Figure 9 for an example) and therefore enables for a time-dependent quantification of potential shading and insolation. To calculate the sun path, we used the software package RayMan (Matzarakis et al., 2007)."*

Furthermore, we newly added Figure 9 to the manuscript, which exemplarily shows the sun path at cluster A and cluster D during summer solstice.

**R2 C14:** *Page 11 lines 6 thru 12: Here the authors describe uncertainties related to freezing conditions and possible salt applied as anti-freeze, e.g. reasons for not using winter-time data, but we never find any cautions about uses of the data they do provide!*

The soil moisture dataset does not contain data for freezing conditions. However, the permittivity dataset contains data for these periods. In the manuscript, we emphasized that a suited model is required when deriving soil moisture from permittivity during those periods (see answer to R1 C3).

The other point concerns the usage of salt as an anti-freeze, which affects electromagnetic soil moisture measurements. We added the following remark to the manuscript: *"Hence, care should be taken when analyzing winter data of $\theta$."*

**R2 C15:** *P11, L25-26: Characterization as vegetated, restricted or free. But, according to Table 2, they only analyzed 3 vegetated sites and 4 restricted drainage sites. Given many other sources of spatial variability and uncertainty, can the authors provide any quantitative basis that we should accept these categorizations?*

Indeed, there is only a small number of plots for the categories "vegetated" and "PPs with restricted drainage". The small number of vegetated sites is due to the fact that only four of the clusters were in the close vicinity of urban green spaces.

The quantification into the categories free drainage and restricted drainage is based on a combination between a visual classification and an analysis of the empirical frequency distribution of soil moisture records during rain events. As illustrated in Fig. 7 the mode of these frequency distributions can be used as a quantitative measure for this classification. To clarify this, we added the following paragraph to the manuscript: *"…
[revised manuscript text omitted]